# Skyrmion dynamics in a frustrated ferromagnetic film and current-induced helicity locking-unlocking transition

Xichao Zhang [1], Jing Xia [1], Yan Zhou[1], Xiaoxi Liu[2], Han Zhang [3] & Motohiko Ezawa [4]

The helicity-orbital coupling is an intriguing feature of magnetic skyrmions in frustrated magnets. Here we explore the skyrmion dynamics in a frustrated magnet based on the $J_1$-$J_2$-$J_3$ classical Heisenberg model explicitly by including the dipole-dipole interaction. The skyrmion energy acquires a helicity dependence due to the dipole-dipole interaction, resulting in the current-induced translational motion with a fixed helicity. The lowest-energy states are the degenerate Bloch-type states, which can be used for building the binary memory. By increasing the driving current, the helicity locking-unlocking transition occurs, where the translational motion changes to the rotational motion. Furthermore, we demonstrate that two skyrmions can spontaneously form a bound state. The separation of the bound state forced by a driving current is also studied. In addition, we show the annihilation of a pair of skyrmion and antiskyrmion. Our results reveal the distinctive frustrated skyrmions may enable viable applications in topological magnetism.

[1] School of Science and Engineering, The Chinese University of Hong Kong, Shenzhen 518172, China. [2] Department of Electrical and Computer Engineering, Shinshu University, 4-17-1 Wakasato, Nagano 380-8553, Japan. [3] SZU-NUS Collaborative Innovation Center for Optoelectronic Science and Technology, Key Laboratory of Optoelectronic Devices and Systems of Ministry of Education and Guangdong Province, College of Optoelectronic Engineering, Shenzhen University, Shenzhen 518060, China. [4] Department of Applied Physics, The University of Tokyo, 7-3-1 Hongo, Tokyo 113-8656, Japan. Xichao Zhang and Jing Xia contributed eqully to this work. Correspondence and requests for materials should be addressed to Y.Z. (email: zhouyan@cuhk.edu.cn) or to M.E. (email: ezawa@ap.t.u-tokyo.ac.jp)

The magnetic skyrmion is an exotic and versatile topological object in condensed matter physics[1–5], which promises novel applications in electronic and spintronic devices[6–9]. It was first experimentally identified in chiral magnetic materials in 2009[10]. Subsequently, skyrmions have been experimentally observed, created, and manipulated in a number of material systems, including magnetic materials[10–21], multiferroic materials[22], ferroelectric materials[23], and semiconductors[24]. Due to the properties of the topologically protected stability as well as the efficient mobility driven by external forces, skyrmions are anticipated to be predominantly employed as information carriers in future data storage devices[25–32], logic computing devices[33], microwave devices[34,35], spin-wave devices[36], and transistor-like devices[37].

Very recently, it was discovered experimentally that the skyrmion lattice can be stabilized by an order-from-disorder mechanism in the triangular spin model with competing interactions[38]. Then, a rich phase diagram of an anisotropic frustrated magnet and properties of frustrated skyrmions with arbitrary vorticity and helicity were investigated[39]. Other remarkable physical properties of skyrmions in the frustrated magnetic system have also been studied theoretically[40–49]. Skyrmions in frustrated magnets are stabilized by the quartic differential term, which is a reminiscence of the original mechanism of the dynamical stabilization proposed by Skyrme[50]. A prominent property is that a skyrmion and an antiskyrmion have the same energy irrespective to the helicity. For instances, the high-topological-number skyrmion[51] and the antiskyrmion[52] with the topological number (i.e., the skyrmion number) of $\pm 2$ are stable or metastable in the anisotropic frustrated magnet[39,41], where the skyrmion shows coupled dynamics of the helicity and the center of mass[39]. It is found that the translational motion of a skyrmion is coupled with its helicity in the frustrated magnet, resulting in the rotational skyrmion motion[41]. These novel properties due to helicity-orbital coupling are lacking in the conventional ferromagnetic system, where the skyrmion is stabilized by the Dzyaloshinskii-Moriya interaction and the helicity is locked.

In this paper, we explore skyrmions and antiskyrmions in a two-dimensional (2D) frustrated ferromagnetic system with competing exchange interactions based on the $J_1$-$J_2$-$J_3$ classical Heisenberg model on a simple square lattice[41]. We explicitly include the dipole-dipole interaction (DDI), which is neglected in the previous literature[39–41]. Our key observation is that the DDI plays an essential role in the frustrated skyrmion physics.

We first study the relaxed spin structure and the energy of a skyrmion with different initial values of the skyrmion number and the helicity. In the framework of our model [cf. Eq. (1)], the energies of skyrmions with different helicities are degenerate in the absence of the DDI, but the degeneracy is resolved by the DDI. A skyrmion has two degenerate lowest-energy states, which are the Bloch-type states with the helicities $\eta = \pm \pi/2$. Namely, a Bloch-type skyrmion has an internal degree of freedom taking a binary value of $\eta = \pm \pi/2$. These two Bloch-type skyrmions are clearly distinguishable since they move straight in opposite directions under a weak driving current, which can result in the skyrmion Hall effect[19,20] where skyrmions with different $\eta$ are accumulated at opposite sample edges. On the other hand, when a strong driving current is applied, the helicity is no longer locked, and the skyrmion performs a rotational motion along with a helicity rotation. We present an effective Thiele equation to account for this helicity locking-unlocking transition induced by the driving current. We also show that it is possible to design a binary memory with the use of the binary helicity state of the Bloch-type skyrmion.

Furthermore, we demonstrate a spontaneous formation of the two-skyrmion bound state (i.e., the bi-skyrmion) as well as the two-antiskyrmion bound state (i.e., the bi-antiskyrmion) due to the attraction between two skyrmions and two antiskyrmions, respectively. Indeed, it is possible to split a bi-skyrmion as well as a bi-antiskyrmion by applying a driving current. The reason is that the direction of the skyrmion motion depends on its helicity. In addition, we demonstrate a pair annihilation of a skyrmion and an antiskyrmion, which emits a propagating spin wave upon the annihilation event. Our results indicate that there are more promising properties and degrees of freedom of skyrmions and antiskyrmions in the frustrated magnetic system, which have great potential to be used in future spintronic and topological applications.

## Results

**Theoretical model and simulations.** We consider the $J_1$-$J_2$-$J_3$ classical Heisenberg model on a simple square lattice[41]. The Hamiltonian can be expressed as

$$\mathcal{H} = -J_1 \sum_{\langle i,j \rangle} \mathbf{m}_i \cdot \mathbf{m}_j - J_2 \sum_{\langle\langle i,j \rangle\rangle} \mathbf{m}_i \cdot \mathbf{m}_j - J_3 \sum_{\langle\langle\langle i,j \rangle\rangle\rangle} \mathbf{m}_i \cdot \mathbf{m}_j$$
$$- H_z \sum_i m_i^z - K \sum_i \left(m_i^z\right)^2 + H_{\text{DDI}}, \quad (1)$$

where $\mathbf{m}_i$ represents the normalized spin at the site $i$, $|\mathbf{m}_i| = 1$. $\langle i,j \rangle$, $\langle\langle i,j \rangle\rangle$, and $\langle\langle\langle i,j \rangle\rangle\rangle$ run over all the nearest-neighbor (NN), next-NN (NNN), and next-NNN (NNNN) sites in the magnetic layer, respectively. $J_1$, $J_2$, and $J_3$ are the coefficients for the NN, NNN, and NNNN exchange interactions, respectively. $H_z$ is the magnetic (Zeeman) field applied along the +$z$-direction, $K$ is the perpendicular magnetic anisotropy constant, $H_{\text{DDI}}$ represents the DDI, i.e., the demagnetization. The total energy of the given system contains the NN exchange energy, the NNN exchange energy, the NNNN exchange energy, the anisotropy energy, the Zeeman energy, and the demagnetization energy.

The simulation is carried out with the 1.2a5 release of the Object Oriented MicroMagnetic Framework (OOMMF) software with the open boundary condition (OBC)[53]. The standard OOMMF extensible solver (OXS) objects are employed, including the OXS object for the calculation of the NN exchange interaction. In addition, we have developed the OXS extension modules for the calculation of the NNN and NNNN exchange interactions, which are implemented in our simulation of the $J_1$-$J_2$-$J_3$ classical Heisenberg model.

The time-dependent spin dynamics in the simulation is described by the Landau-Lifshitz-Gilbert (LLG) equation[53]

$$\frac{d\mathbf{m}}{dt} = -\gamma_0 \mathbf{m} \times \mathbf{h}_{\text{eff}} + \alpha \left(\mathbf{m} \times \frac{d\mathbf{m}}{dt}\right), \quad (2)$$

where $\mathbf{h}_{\text{eff}} = -\delta\mathcal{H}/\delta\mathbf{m}$ is the effective field, and $\gamma_0$ is the absolute gyromagnetic ratio. For the simulation including a vertical spin current generated by the spin Hall effect, the Slonczewski-like spin-transfer torque (STT)

$$\tau = -u\mathbf{m} \times (\mathbf{m} \times \mathbf{p}) \quad (3)$$

will be added to the right-hand side of equation (2) with

$$u = \frac{\gamma_0 \hbar j P}{2ae\mu_0 M_S}, \quad (4)$$

where $\hbar$ is the reduced Planck constant, $\mu_0$ is the vacuum permeability constant, $e$ is the electron charge, $j$ is the current density, $P = 0.4$ is the spin Hall angle, $a = 4$ Å is the lattice constant, $M_S$ is the saturation magnetization, and $\mathbf{p} = -\hat{y}$ is the spin-current polarization direction. Besides, we also employed the OOMMF conjugate gradient (CG) minimizer for the spin

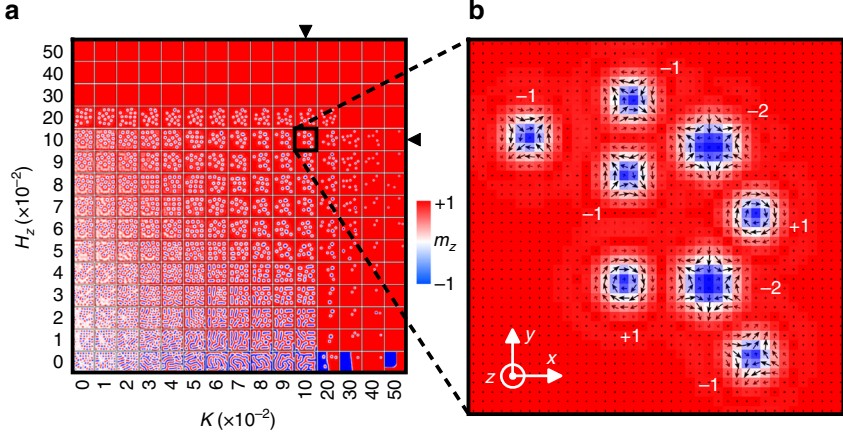

**Fig. 1** Skyrmions and antiskyrmions in a frustrated $J_1$-$J_2$-$J_3$ ferromagnetic thin film. **a** Typical metastable states obtained by relaxing the magnetic thin film with a random initial spin configuration. **b** The spin configuration of the relaxed sample with $K = H_z = 0.1$. The skyrmion number $Q$ is indicated. The arrow denotes the in-plane spin components ($m_x$, $m_y$). The color scale represents the out-of-plane spin component ($m_z$), which has been used throughout this paper. The model is a square element (40 × 40 spins) with the OBC. The fixed parameters (in units of $J_1 = 1$) are $J_2 = -0.8$ and $J_3 = -1.2$. $K$ and $H_z$ are varied between 0 and 0.5, respectively

relaxation simulation, which locates local minima in the energy surface through direct minimization techniques[53].

In this work, the default values for the NN, NNN, and NNNN exchange interactions are given as $J_1 = 3$ meV, $J_2 = -0.8J_1$, and $J_3 = -1.2J_1$, respectively. $K = 0$–$0.5J_1$, $H_z = 0$–$0.5J_1$, $\alpha = 0.1$, $\gamma_0 = 2.211 \times 10^5$ m A$^{-1}$ s$^{-1}$, and $M_S = 800$ kA m$^{-1}$. The geometry of the cubic cells generated by the spatial discretization is 4 Å × 4 Å × 4 Å.

We define the skyrmion number of a spin texture in the continuum limit by the formula

$$Q = -\frac{1}{4\pi} \int d^2\mathbf{r} \cdot \mathbf{m}(\mathbf{r}) \cdot (\partial_x \mathbf{m}(\mathbf{r}) \times \partial_y \mathbf{m}(\mathbf{r})). \quad (5)$$

The normalized spin field $\mathbf{m}(\mathbf{r})$ takes a value on the 2-sphere $S^2$. The skyrmion number $Q$ counts how many times $\mathbf{m}(\mathbf{r})$ wraps $S^2$ as the coordinate $(x, y)$ spans the whole planar space. We parametrize the spin texture as

$$\mathbf{m}(\mathbf{r}) = \mathbf{m}(\theta, \phi) = (\sin\theta\cos\phi, \sin\theta\sin\phi, \cos\theta), \quad (6)$$

with

$$\phi = Q\varphi + \eta, \quad (7)$$

where $\varphi$ is the azimuthal angle ($0 \leq \varphi < 2\pi$) and $\eta$ is the helicity defined mod $2\pi$. Namely, the helicity $\eta$ and $\eta - 2\pi$ are identical. Special values of the helicity play important roles, that is, $\eta = 0$, $\pi/2$, $\pi$, $3\pi/2$. We frequently use $\eta = -\pi/2$ instead of $\eta = 3\pi/2$. We may index a skyrmion as $(Q, \eta)$ with the use of the skyrmion number and the helicity. A skyrmion with $Q < 0$ may also be called an antiskyrmion.

**Typical metastable states**. We first relax the frustrated magnetic system from a random initial spin configuration for different values of $K$ and $H_z$. The initial spin configuration is random in three dimensions. Figure 1a shows the typical metastable states in the relaxed sample with respect to $K$ and $H_z$ solved by the OOMMF CG minimizer (cf. Supplementary Fig. 1). The same typical metastable states obtained by the OOMMF solver integrating the LLG equation is given in Supplementary Fig. 2, which shows qualitatively identical results, justifying the reliability of the relaxation procedure by the CG minimizer. It shows that isolated

skyrmions and antiskyrmions can be spontaneously formed in the relaxed state where the magnetic system has certain values of $K$ and $H_z$. Small values of $K$ and $H_z$ lead to the formations of distorted skyrmions and antiskyrmions, stripe domains, and large in-plane spin configurations, where the typical size of the undistorted skyrmions with $Q = 1$ and stripe domains is about five times the lattice constant. Figure 1b shows the detailed spin configuration of the relaxed sample with $K = 0.1$ and $H_z = 0.1$ (in units of $J_1 = 1$). It can be seen that skyrmions with $(+1, -\pi/2)$, antiskyrmions with $(-1, \pi/2)$, $(-1, -\pi/2)$, and $(-2, \pi/2)$, coexist in the relaxed sample. It is noteworthy that the high-topological-number antiskyrmion with $Q = -2$ can be seen as a two-antiskyrmion bound state or a bi-antiskyrmion, about which we will discuss later.

The existence of skyrmions and antiskyrmions also depends on the relation between $J_1$, $J_2$, and $J_3$ (cf. Supplementary Figs. 3–6), however, the presence of the DDI has limited influence on the existence of skyrmions and antiskyrmions as the typical metastable states (cf. Supplementary Figs. 7–10). Besides, as our simulations are carried out with the OBC in the presence of the DDI, the orientations of spins at the edge could be different from that in the bulk at certain conditions (cf. Supplementary Fig. 11). For the same reason, the spin textures, such as the stripe domains, obtained in our typical metastable states are not periodic patterns. The lack of regular lattice structure of skyrmions has been observed in several recent experiments[15,18–21].

**Stability of isolated skyrmions and antiskyrmions**. We study possible static solutions of isolated skyrmions and antiskyrmions by relaxing the sample with a given initial spin configuration. Namely, we construct a number of possible standard skyrmion and antiskyrmion spin structures with different values of $(Q, \eta)$ as the initial spin configuration of the sample. After relaxation of the system, we obtain the relaxed states of skyrmions and anti-skyrmions, which are stable or metastable solutions. As shown in Fig. 2, we construct a radial symmetric skyrmion as the initial state. We check whether the initial geometry is preserved or destroyed after the relaxation. For the initial states of skyrmions with $Q = \pm 1$, $\pm 2$, $\pm 3$, the numbers $(Q, \eta)$ do not change after the relaxation although the radius shrinks. However, the skyrmion with $Q = \pm 4$ is unstable and splits into two skyrmions with $Q = \pm 2$. The skyrmion with $Q = \pm 5$ is also unstable and splits into five skyrmions with $Q = \pm 1$. The results of the antiskyrmion are

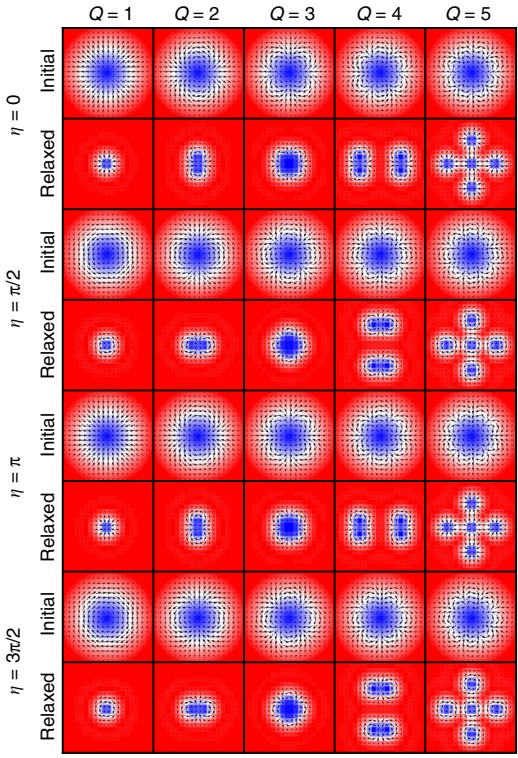

**Fig. 2** Relaxed states of skyrmions with different values of $Q$ and $\eta$. We construct a radial symmetric skyrmion as the initial state and check whether the initial geometry is preserved or destroyed after the relaxation. The model is a square element (40 × 40 spins) with the OBC. The parameters are $J_2 = -0.8$, $J_3 = -1.2$, $K = 0.1$, and $H_z = 0.1$

similar to those of the skyrmion as shown in Supplementary Fig. 12.

**Binary stable skyrmions and antiskyrmions**. The skyrmion energy and the antiskyrmion energy are degenerate and independent of the helicity $\eta$ in the absence of the DDI (cf. ref. [41]). We also present a symmetric analysis of this property in Supplementary Note 1. Needless to say, the DDI exists in all magnetic materials, however, it has been customarily considered negligible for a microscopic spin texture. This is actually not the case. We show the detail of each contribution to the skyrmion energy in Fig. 3, where the DDI energy is found to be comparable to the anisotropy or Zeeman energy. Indeed, the contribution of the DDI energy is related to the material and geometric parameters of the given sample. We show in Supplementary Fig. 13 that the DDI energy decreases with increasing $M_S$ and increasing thickness of the sample. The DDI energy is independent of the length-to-width ratio, but the DDI energy to total skyrmion energy ratio slightly decreases with increasing thickness. Thus, it is expected that the DDI effect will be more significant in thick samples with large $M_S$. Also, the NN, NNN, and NNNN exchange interaction energies are independent of the length-to-width ratio, but significantly varies with the thickness.

As indicated in Fig. 4, there are two effects from the DDI. First, the degeneracy is resolved between the skyrmion and the antiskyrmion. Second, the degeneracy is resolved between various helicities in general. The Bloch-type (Néel-type) skyrmions with $\eta = \pm\pi/2$ ($\eta = 0, \pi$) are found to possess the minimum energy for $Q = \pm 1$ ($Q = \pm 3$). It is consistent with the fact that the DDI is due to the magnetic charge $\rho_{mag} = \nabla \cdot \mathbf{m}$ and prefers the Bloch-type structure[5]. On the other hand, the energy is independent of the helicity for $Q = \pm 2$. The helicity dependence

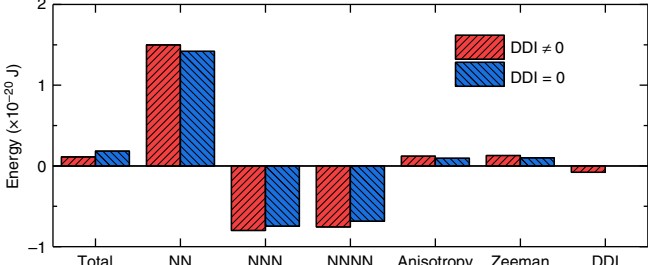

**Fig. 3** Details of the micromagnetic energy of a relaxed (1, $\pi$/2)-skyrmion in the presence and absence of the DDI. The contribution from the DDI energy is found to be comparable to the anisotropy or Zeeman energy. The skyrmion energy is determined by the energy difference between the sample with a skyrmion and the sample without a skyrmion (i.e., with the FM state)

of the energy of antiskyrmions is similar but weaker than that of skyrmions, as shown in Fig. 4b. This is because an antiskyrmion has the antivortex-like structure, where the contribution of the DDI is smaller than the vortex structure of a skyrmion. The helicity dependence of the energy is a new feature due to the DDI, which is overlooked in the previous literature[41].

As a result, all skyrmions ($Q = \pm 1$) with any $\eta$ relax to the Bloch-type skyrmions with $\eta = \pm\pi/2$ since they have the minimum energy. Exceptions are those possessing the precise initial values of $\eta = 0, \pi$. Indeed, we have studied the relaxation of a skyrmion with ($\pm 1, \eta$) for different initial values of $\eta$ in the range of $\eta = 0-2\pi$. As shown in Fig. 5, the skyrmions with the initial helicity of $\eta = 0, \pi/2, \pi$, and $3\pi/2$ keep their helicity during the relaxation. However, for the skyrmions with other initial helicities, the helicity is relaxed to $\eta = \pm\pi/2$, justifying that the most stable skyrmions have $\eta = \pm\pi/2$. Consequently, the stable skyrmion is the Bloch-type skyrmion carrying the internal degree of freedom indexed by a binary value ($\eta = \pm\pi/2$).

**Current-induced dynamics**. A skyrmion and an antiskyrmion move rotationally together with a helicity rotation in the frustrated system without the DDI[41], as shown in Fig. 6a, b. The skyrmion and antiskyrmion motions drastically change once the DDI is introduced. Indeed, the DDI produces the energy difference depending on the helicity as shown in Fig. 4. Namely, the helicity rotation costs some energies in the order of $0.01 \times 10^{-20}$ J. Figure 5 shows the time evolution of the helicity. The helicity rotates toward that of the Bloch-type state ($\pm\pi/2$) unless the initial helicity is precisely of the Néel-type state.

This helicity locking occurs for the small driving current, since the Bloch-type state has the lowest energy in the presence of the DDI. Namely, under the small current injection, the helicity relaxes to that of the Bloch-type state, and then the skyrmion or antiskyrmion moves along a straight line with the helicity fixed, as shown in Figs. 6c and d. This translational motion occurs due to the helicity locking by the DDI, which is highly contrasted to the case without the DDI.

However, as shown in Figs. 6e and f, the translational motion with the fixed helicity evolves to be a rotational motion together with a helicity rotation once the driving current density exceeds a certain critical value $j_c$ (cf. Supplementary Figs. 14–16 and Supplementary Movies 1 and 2). The reason is that the energy injected by the driving current overcomes the potential difference between the Bloch-type and Néel-type skyrmions.

We have developed an effective Thiele equation to account for the helicity locking-unlocking transition (cf. Supplementary Notes 2 and 3). Adding an effective potential term due to the DDI, the effective Thiele equation becomes identical to the forced

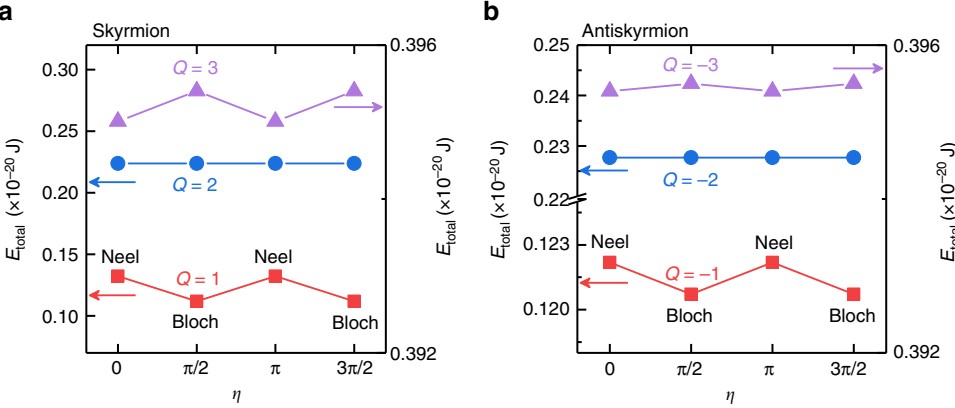

**Fig. 4** Total micromagnetic energy $E_{total}$ for **a** relaxed skyrmions and **b** relaxed antiskyrmions as functions of $Q$ and $\eta$. The lowest energy states are found to be two degenerate Bloch-type states both for the skyrmion ($Q = 1$) and the antiskyrmion ($Q = -1$) due to the DDI. The corresponding spin configurations for relaxed skyrmions and antiskyrmions are given in Fig. 2 and Supplementary Fig. 12, respectively

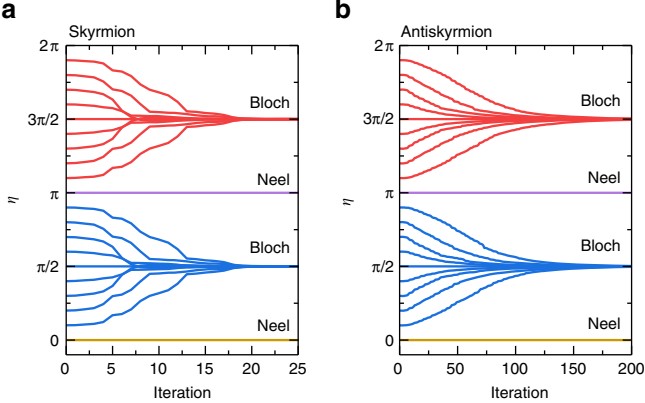

**Fig. 5** Helicity $\eta$ as functions of the iteration for **a** skyrmions and **b** antiskyrmions with varied initial $\eta$. All of them are relaxed to the Bloch-type state unless they are initially in the Néel-type state. The model is a square element (11 × 11 spins) with the OBC. The parameters are $J_2 = -0.8$, $J_3 = -1.2$, $K = 0.1$, and $H_z = 0.1$

oscillation problem of a pendulum in the presence of friction, where the force is provided by the driving current while the friction by the Gilbert damping. It is well known that when the force is less than the threshold the pendulum stops due to the friction, which corresponds to the straight motion of a skyrmion with helicity fixed as in Fig. 6c. On the other hand, when the force is larger than the threshold, the pendulum rotates around the rotational center, which corresponds to the rotational motion of a skyrmion with helicity rotating as in Fig. 6e. The critical value of the driving current density is given by

$$j_c = \frac{4U_0 a e \mu_0 M_S}{\gamma_0 \hbar^2 P_\eta},\qquad(8)$$

where $U_0$ is the potential barrier between Bloch-type and Néel-type skyrmions, and $Y_\eta$ quantifies the efficiency of the driving force acting on the skyrmion (cf. Supplementary Notes 2 and 3). It can be seen that $j_c$ increases with $U_0$ as a larger driving force is required to overcome a stronger barrier between Bloch-type and Néel-type skyrmions for the rotational motion. Also, $j_c$ decreases with increasing $Y_\eta$ because that the magnitude of the driving force provided by the STT for a certain current density is proportional to $Y_\eta$. The values of $U_0$ and $Y_\eta$ depend on the spin configuration of the skyrmion (cf. Figure 4 and Supplementary Notes 2 and 3). For a skyrmion with $\eta = \pm\pi/2$ as shown in Fig. 6c, we numerically

found $j_c \sim 94 \times 10^{10}$ A m$^{-2}$ and $U_0 \sim 0.02 \times 10^{-20}$ J, thus $Y_\eta \sim 5.7$. We note that the spin configuration of the skyrmion is determined by parameters such as $J_2$, $J_3$, $K$, and $H_z$, which means $j_c$ can be adjusted by controlling these parameters in experiments. Indeed, we numerically found that $j_c$ is proportional to the values of $J_2$, $J_3$, $K$, and $H_z$ (cf. Supplementary Fig. 17). Note that the dependences of $j_c$ on $a$, $P$, and $M_S$ are trivial as the magnitude of the driving force is proportional to $jP/aM_S$ [cf. equation (4)].

Additionally, we found that the helicity locking-unlocking transition exists even when the thermal effect (cf. Methods) is included (cf. Supplementary Fig. 18). The trajectory of a skyrmion with locked helicity fluctuates, and the fluctuation increases with temperature. When the helicity is unlocked by the driving current, the skyrmion dynamics becomes rather complex at finite temperatures, which can be seen as a combination of rotational motion and Brownian motion[32].

**Switching process of binary Bloch-type skyrmions**. We have revealed prominent properties of the helicity-orbital coupling in the skyrmion motion in the frustrated magnetic system with the DDI. We may design a binary memory together with a switching process based on these properties. Let us start with a skyrmion at rest. (i) It is in one of the two Bloch-type states indexed by the helicity $\eta = \pm\pi/2$; since the helicity is a conserved quantity, we may use it as a one-bit memory. (ii) We can read its value by observing the motion of a skyrmion; when a small current pulse is applied, a skyrmion begins to move along a straight line toward left or right according to its helicity $+\pi/2$ or $-\pi/2$ [cf. Fig. 6c and Supplementary Movie 3]; this is the read-out process of the memory. (iii) On the other hand, when we apply a strong current pulse, a skyrmion begins to rotate together with its helicity rotation; we can tune the pulse period to stop the helicity rotation so that it is relaxed to the flipped value of the initial helicity, as shown in Fig. 7. This is the switching process of the memory. As shown in Supplementary Movie 4, a skyrmion moving rightward is stopped and moves leftward after one rotation under a strong current pulse applied for an appropriate period.

**Spontaneous formations of a bi-skyrmion and a bi-antiskyrmion**. We proceed to investigate the spontaneous formations of a bi-skyrmion and a bi-antiskyrmion, i.e. the spontaneous formations of a high-topological-number skyrmion and a high-topological-number antiskyrmion. We construct one skyrmion with $\eta = \pi/2$ and one skyrmion with $\eta = -\pi/2$, which are placed at the left and right sides of the sample, respectively, with

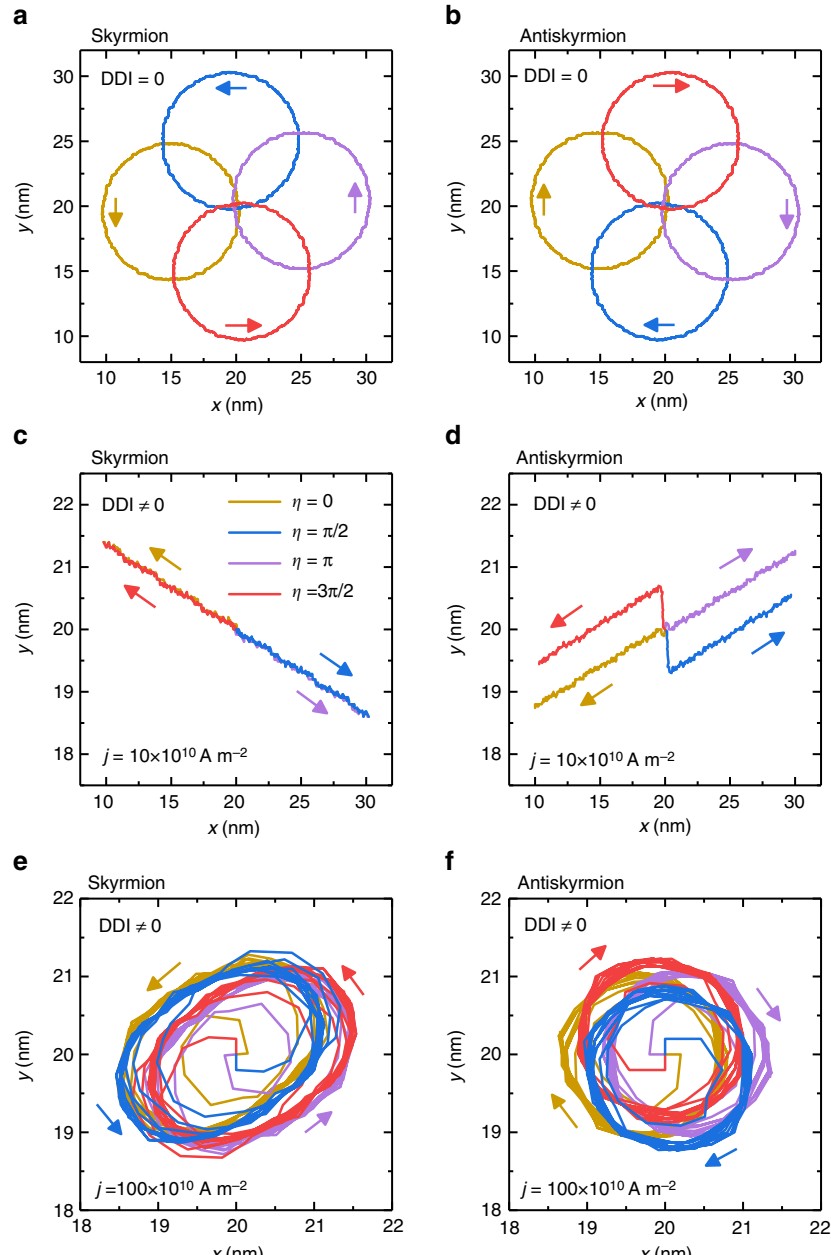

**Fig. 6** Trajectories of skyrmions and antiskyrmions in a frustrated $J_1$-$J_2$-$J_3$ ferromagnetic thin film. Trajectories of **a** skyrmions and **b** antiskyrmions with different initial $\eta$ in the absence of the DDI. Trajectories induced by **c**, **d** a small driving current ($j = 10 \times 10^{10}$ A m$^{-2}$) and by **e**, **f** a large driving current ($j = 100 \times 10^{10}$ A m$^{-2}$) in the presence of the DDI. In the presence of the DDI, the skyrmion motion is along a straight line with the helicity fixed for the small driving current, but it is along a circle together with the helicity rotation for the large driving current. This is the helicity locking-unlocking transition. The model is a square element ($100 \times 100$ spins) with the OBC. The parameters are $J_2 = -0.8$, $J_3 = -1.2$, $K = 0.1$, $H_z = 0.1$, and $\alpha = 0.1$

the initial configuration as shown in Fig. 8. The dynamic process of the spontaneous formation of a bi-skyrmion is illustrated in Fig. 8a, where a time-dependent relaxation of the system is simulated. The two skyrmions first move close to each other, where they both have in-plane spins near each other that are pointing along the +$y$-direction. The two skyrmions then smoothly merge into one bi-skyrmion with $Q = +2$, when the encountered in-plane spins are rotated into the +$z$-direction. Finally, a bi-skyrmion with $Q = +2$ is formed in the sample (cf. Supplementary Movie 5). A similar spontaneous formation of a bi-antiskyrmion with $Q = -2$ is possible as given in Fig. 8b (cf. Supplementary Movie 6). It is interesting that there is a skyrmion-skyrmion (antiskyrmion-antiskyrmion) interaction leading to spontaneous

formation of a bi-skyrmion (bi-antiskyrmion), whose detailed analysis is given in Supplementary Figs. 19–22.

**Forced separations of a bi-skyrmion and a bi-antiskyrmion.** After showing the spontaneous formations of the bi-skyrmion and the bi-antiskyrmion, we further demonstrate that it is possible to split the bi-skyrmion and the bi-antiskyrmion by applying a driving current.

Figure 9 demonstrates the forced separation of two skyrmions from a bi-skyrmion. We first place a relaxed bi-skyrmion at the center of the sample. Then, a driving current of $j = 8 \times 10^{11}$ A m$^{-2}$ is vertically injected to the sample, which can be realized by the

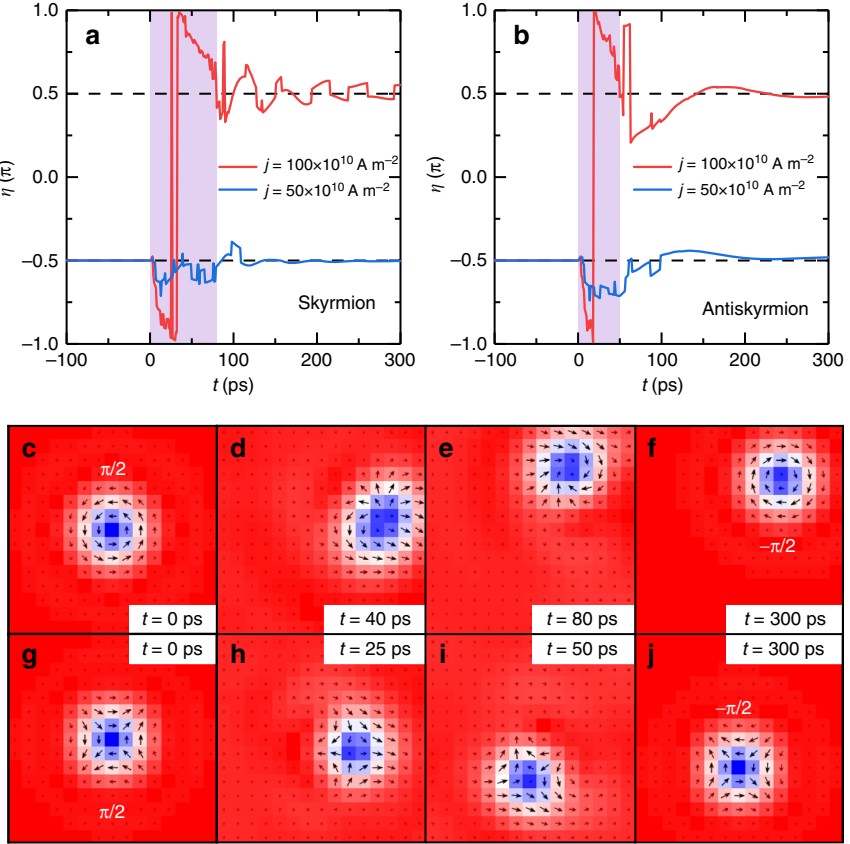

**Fig. 7** Flip of the skyrmion (antiskyrmion) helicity induced by a current pulse. **a** Skyrmion helicity as a function of time. A skyrmion is at rest with the helicity being $\eta = -\pi/2$. The helicity is flipped by a strong 80-ps-long current pulse and becomes $\eta = \pi/2$. Such a flip does not occur by applying a small current pulse with the same duration. **b** A similar flip occurs also for an antiskyrmion. **c-f** Snapshots of the skyrmion during the helicity flip process. **g-j** Snapshots of the antiskyrmion during the helicity flip process. The model is a square element (39 × 39 spins) with the OBC. The parameters are $J_2 = -0.8$, $J_3 = -1.2$, $K = 0.1$, $H_z = 0.1$, and $\alpha = 0.1$

spin Hall effect in a heavy-metal substrate. It is found that the bi-skyrmion starts to rotate counterclockwise, and forms a clear peanut-like shape, as shown in Fig. 9f. When it has rotated almost 180 degrees, two skyrmions are almost generated, as shown in Fig. 9k. As shown in Fig. 9l, the bi-skyrmion is successfully split into two isolated skyrmions (cf. Supplementary Movie 7). This forced separation is possible since the bi-skyrmion is composed of two skyrmions with opposite helicities, which rotate in opposite directions. The separation process of a bi-antiskyrmion forced by the driving current is similar to that of the bi-skyrmion (cf. Supplementary Fig. 23 and Supplementary Movie 8). It is worth mentioning that the separation of a bi-skyrmion may spontaneously occur at a finite temperature. Thermal effects could also result in the collapse and annihilation of a bi-skyrmion (cf. Supplementary Fig. 24).

We note that the total skyrmion number conserves during both the spontaneous formation and the forced separation of the bi-skyrmion and the bi-antiskyrmion. Furthermore, we found that the bound state with a higher skyrmion number $|Q| > 2$ can also be spontaneously generated (cf. Supplementary Figs. 25 and 26).

**Pair annihilation of a skyrmion and an antiskyrmion.** Finally, we demonstrate the pair annihilation process of a skyrmion with an antiskyrmion in a forced collision event driven by the current, as shown in Fig. 10. We first place a relaxed skyrmion and a relaxed antiskyrmion at the center of the sample with a certain spacing between the skyrmion and the antiskyrmion. Then, a 200-ps-long driving current pulse of $j = 8 \times 10^{11}\,\mathrm{A\,m^{-2}}$ is

vertically injected to the sample. The skyrmion and the antiskyrmion move in a circular trajectory toward opposite directions. The helicity $\eta$ of the skyrmion and the antiskyrmion varies with the position as it is coupled with the orbital motion (cf. Fig. 6). As shown in Fig. 10b, the skyrmion meets the antiskyrmion, where the skyrmion has the helicity of $\eta = \pi/2$ while the antiskyrmion has the helicity of $\eta = -\pi/2$, which is the condition for the pair annihilation. The skyrmion and the antiskyrmion then form a magnetic bubble state with $Q = 0$, as shown in Fig. 10d. The magnetic bubble state, which is not topologically protected, rapidly shrinks in size and is finally evolved into the trivial ground state, as shown in Fig. 10l (cf. Supplementary Movie 9). A forced collision between a skyrmion and an antiskyrmion will lead to the pair annihilation of the skyrmion and the antiskyrmion as long as they do not have identical $\eta$ (cf. Supplementary Fig. 8). It is noteworthy that a propagating spin wave is emitted during the pair annihilation event, which can be discerned from the propagating skyrmion number density wave in Fig. 10.

**Discussion**
In this paper, we have studied magnetic skyrmions in a 2D frustrated magnetic system with competing exchange interactions based on the $J_1$-$J_2$-$J_3$ classical Heisenberg model on a simple square lattice. We have demonstrated that the contribution of the DDI energy is significant although the scale of a skyrmion is of the order of nanometer. As a result, the Bloch-type skyrmion has

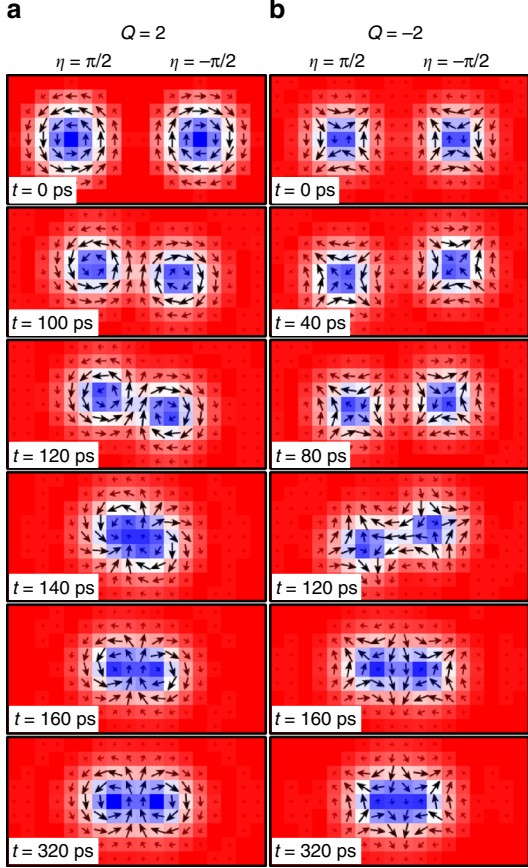

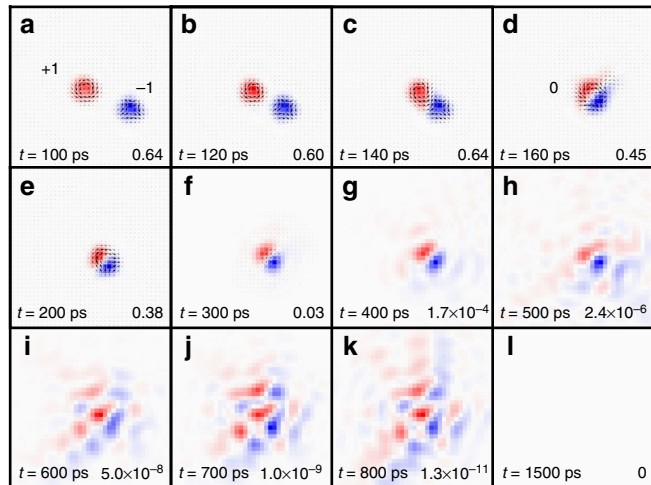

**Fig. 10** Pair annihilation of a skyrmion and an antiskyrmion. **a** We first place a skyrmion and an antiskyrmion. **d** They are spontaneously combined into a magnetic bubble ($Q = 0$). Then, by emitting spin waves, it disappears. The spin wave can be discerned from **g**–**k**. The positive and negative skyrmion number densities are denoted by red and blue colors, respectively. $Q$ is indicated in **a** and **d**. The maximum value of the $Q$ density is indicated in the lower right corner of each snapshot. The model is a square element (60 × 60 spins) with the OBC. The parameters are $J_2 = -0.8$, $J_3 = -1.2$, $K = 0.1$, $H_z = 0.1$, $\alpha = 0.1$, and $j = 8 \times 10^{11}$ A m$^{-2}$

**Fig. 8** Spontaneous formations of a bi-skyrmion and a bi-antiskyrmion. **a** Spontaneous formation of a bi-skyrmion by merging two skyrmions with different $\eta$. **b** Spontaneous formation of a bi-antiskyrmion by merging two antiskyrmions with different $\eta$. The model is a square element (18 × 9 spins) with the OBC. The parameters are $J_2 = -0.8$, $J_3 = -1.2$, $K = 0.1$, $H_z = 0.1$, and $\alpha = 0.1$

the lowest energy. It has a bi-stable structure indexed by the helicity $\eta = \pm\pi/2$.

The role of the helicity-orbital coupling becomes remarkable in the presence of the DDI. First, a skyrmion moves along a straight line with a fixed helicity under a weak driving current. The moving direction of a skyrmion is opposite when its helicity is opposite. Second, the dynamics of a skyrmion drastically change from the translational motion to the circular motion with the increase of the driving current, which is the locking-unlocking transition of the helicity. Third, we can flip the helicity of a skyrmion by applying a strong current pulse with a reasonable period. We have argued that these properties could be used to design a binary memory together with a read-out process as well as the switching process.

We have also shown that magnetic skyrmions with $Q = \pm2, \pm3$ are possible metastable topological non-trivial states in the frustrated magnetic system. The skyrmion with a large skyrmion number $|Q| > 1$ could be seen as a bound state or a cluster of skyrmions, which has been observed in several recent experiments[54,55]. In particular, we have demonstrated the spontaneous formation and the forced separation of a bi-skyrmion. In addition, we have shown the pair annihilation of a skyrmion and an antiskyrmion by triggering a collision event.

We have explicitly employed a square lattice model in order to investigate skyrmion physics numerically in the frustrated magnetic system. However, the concept of skyrmions as well as the LLG equation are valid for any other lattice systems. In particular, triangular magnets are experimentally feasible as we will discuss below. The degeneracy among different helical states may also be resolved by the in-plane quartic magnetic anisotropy, provided that it is allowed by the symmetry of the underlying spin lattice. An example is known in a square lattice model[56]. However, this anisotropy usually exists in cobalt ferrite, and is negligible in ordinary magnets.

Although there is already a theoretical report on skyrmion lattices[38], isolated skyrmions are yet to be observed in frustrated spin systems. Candidates for these systems would be triangular magnets with transition mental ions (cf. ref. 39), such as

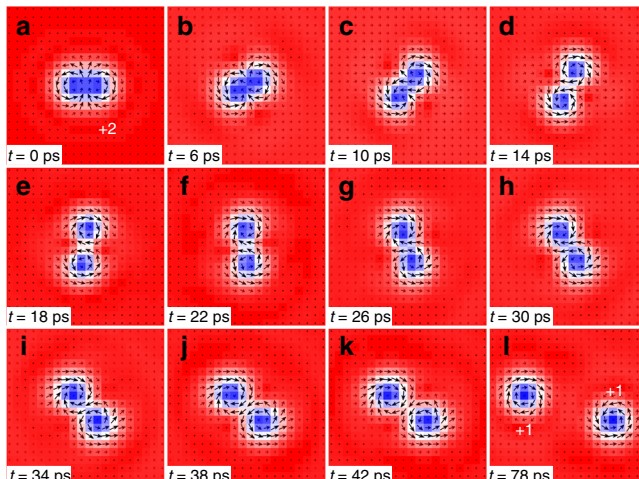

**Fig. 9** Forced separation of two skyrmions from a bi-skyrmion with $Q = 2$. **a** We first place a relaxed bi-skyrmion. **b** It starts to rotate counterclockwise under an injected driving current. After forming a clear peanut-like shape, it is separated into two skyrmions as in **l**. The model is a square element (40 × 40 spins) with the OBC. The parameters are $J_2 = -0.8$, $J_3 = -1.2$, $K = 0.1$, $H_z = 0.1$, $\alpha = 0.1$, and $j = 8 \times 10^{11}$ A m$^{-2}$. $Q$ is indicated in **a** and **l**

$NiGa_2S_4$[57,58], $\alpha$-$NaFeO_2$[59] and dihalides $Fe_xNi_{1-x}Br_2$[60]. It is worth mentioning that in $NiGa_2S_4$[58], a large value of $|J_3/J_1| \approx 3$ is reliable, which could be the condition for stabilizing the frustrated skyrmion. At certain values of $J_1$, $K$, and $H_z$, the values of $J_2$ and $J_3$ required for stabilizing frustrated skyrmions can be varied in a wide range (cf. Supplementary Fig. 6). However, in case only a small magnitude of $J_3$ is reachable, a relatively large magnitude of $J_2$ will be required for stabilizing the frustrated skyrmion, indicating the importance of the antiferromagnetic exchange interactions on the stabilization of the non-collinear spin texture in a ferromagnetic background. Moreover, the anisotropy of dihalides can be adjusted by chemical substitutions, where $Fe_xNi_{1-x}Br_2$ has an easy-axis anisotropy along the $c$ axis for $x > 0.1$[61].

Various properties of skyrmions can be investigated experimentally by using similar techniques elaborated for ferromagnetic systems. The center-of-mass dynamics of a skyrmion can be observed by using time-resolved X-ray microscopy[18,20,21] or the topological Hall effect[62,63]. The helicity of a skyrmion can be directly observed by Lorentz transmission electron microscopy (LTEM) for both skyrmions[14,64,65] and antiskyrmions[66]. The helicity can also be observed by the differential phase contrast scanning transmission electron microscopy (DPC STEM)[67,68]. Besides, a new experimental protocol for the identification of topological magnetic structures with different helicities, by soft X-ray spectroscopy, has been proposed[69]. We also note that a measurable quantity associated with the skyrmion helicity is the toroidal moment[70].

Our theoretical results have revealed the exotic and promising properties of skyrmions and antiskyrmions in frustrated magnetic systems, which may lead to novel spintronic and topological applications based on the manipulation of skyrmions and antiskyrmions.

## Methods

**Thermal effect.** The magnetization dynamics including the thermal effect is described by the stochastic LLG equation[32], in which the effective field reads

$$\mathbf{h}_{\text{eff}} = -\frac{\delta \mathcal{H}}{\delta \mathbf{m}} + \mathbf{h}_{\text{f}}. \tag{9}$$

$\mathbf{h}_{\text{f}}$ is a highly irregular fluctuating field representing the irregular influence of temperature, which is generated from a Gaussian stochastic process and satisfies

$$<h_i(\mathbf{x}, t)> = 0, <h_i(\mathbf{x}, t)h_j(\mathbf{x}', t')> = \frac{2\alpha k_B T}{\hbar} a^2 \delta_{ij}\delta(\mathbf{x} - \mathbf{x}')\delta(t - t'), \tag{10}$$

where $i$ and $j$ are Cartesian components, $a^2$ is the area of lattice, and $\delta_{ij}$ and $\delta(\ldots)$ stand for the Kronecker and Dirac delta symbols, respectively. The finite-temperature simulations are performed with a fixed time step of $5 \times 10^{-15}$ s, while the time step in zero-temperature simulations is adaptive.

**Code availability**. The micromagnetic simulator OOMMF used in this work is publicly accessible at http://math.nist.gov/oommf.

**Data availability**. The data that support the findings of this study are available from the corresponding authors upon reasonable request.

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

## Acknowledgements

X.Z. was supported by JSPS RONPAKU (Dissertation Ph.D.) Program. X.Z. would like to thank Wanjun Jiang and Shi-Zeng Lin for fruitful discussions on frustrated skyrmions. Y.Z. acknowledges the support by the President's Fund of CUHKSZ, the National Natural Science Foundation of China (Grant No. 11574137), and Shenzhen Fundamental Research Fund (Grant Nos. JCYJ20160331164412545 and JCYJ20170410171958839). M.E. acknowledges the support by the Grants-in-Aid for Scientific Research from JSPS KAKENHI (Grant Nos. 25400317, JP17K05490 and JP15H05854), and also the support by CREST, JST (JPMJCR16F1). M.E. is very much grateful to Naoto Nagaosa for many helpful discussions on the subject.

## Author contributions

M.E. and Y.Z. conceived the idea. Y.Z., M.E., and X.L. coordinated the project. X.Z. performed the numerical simulation and data analysis. J.X. developed the modules for NNN and NNNN exchange interactions. M.E. carried out the theoretical analysis. X.Z. and M.E. drafted the manuscript and revised it with input from Y.Z., X.L. and H.Z. All authors discussed the results and reviewed the manuscript. X.Z. and J.X. contributed equally to this work.

## Additional information

**Competing interests:** The authors declare no competing financial interests.

