## [Peer Review File · Nature Communications]

Reviewers' comments:

Reviewer #1 (Remarks to the Author):

Referee Report

Skyrmions and Antiskyrmions in a Frustrated J1-J2-Jr Ferromagnetic Film:
Current-induced Helicity Locking-Unlocking Transitions.

Z. Zhang et al

This paper is a theoretical-computational study of skyrmion states in a frustrated magnet. Skyrmions are a subject of intense interest and have been found in numerous chiral magnets. Frustrated systems are another place where such skyrmions may arise as has been reported by other authors including some on this paper. Here the authors consider the effect of dipole-dipole interactions and examine which gives rise to a helicity. The authors use micromagnetic simulations to show that skyrmion states with different Q values can arise and then study various aspects of dynamics such as helicity locking and unlocking, annihilation and creation of bound states. They indicate how the dynamics of such states could be valuable for the creation of new kinds of skyrmion devices.

Overall the results are interesting and the dynamics of skyrmions in frustrated magnetism is not very well known. The authors find a rich variety of dynamical behaviors. I have some questions before making a conclusion either way.

(1) The authors should go over the text as some of the wording is bit awkward; however, the paper is in general easy to follow.

(2) The authors do not spend much time on thermal effects. Thermal effects are known to be important for stabilizing skyrmions in chiral magnets, but how about in frustrated magnets? Additionally, could one have thermal unbinding of bound pairs?

(3) Experimentally what would one look for in these systems? For example, what signatures could be observed through direct imaging, transport or neutron scattering?

(4) The skyrmion lattice structure generally looks disordered in these systems. Is this due to the frustrating effects or the equilibration time in the simulations, or is this a general feature one would expect in these types of skyrmions as opposed to skyrmions in DMI materials?

(5) The authors discuss the dynamics; however, experimentally how would one detect this? Could one look at changes in the topological Hall effect?

(6) What would be the typical lattice constant of these systems or the size of the skyrmions? Do the authors have any particular materials in mind or do experimentalists need to find a way to make J1-J3 materials in the right conditions to find these states?

Reviewer #2 (Remarks to the Author):

"Skyrmions and Antiskyrmions in a Frustrated J1-J2-J3 Ferromagnetic Film: Current-induced Helicity Locking-Unlocking Transition" by X. Zhang et al.

The authors discuss static and dynamic properties of skyrmions in a frustrated square-lattice magnet with competing exchange interactions and long-range magnetodipolar interactions. The dipole-dipole interaction favors Bloch-type skyrmions (helcity angle equal plus/minus $\pi/2$), as is also the case for magnetic bubbles with skyrmion topology, and the authors explore the effect of this "helcity fixing" on the current-driven dynamics of skyrmions. They find, in particular, that the direction of motion excited by low electric currents depends on the skyrmion helicity, while stronger currents give rise to helicity rotation accompanying the translational motion of skyrmion. The latter effect is easy to understand, but the former one is a surprise to me. The authors suggest that these effects can be used to make new magnetic memory devices.

I have mixed feelings about this work. While some results look interesting, I am not happy with the overall quality of the manuscript. As is elaborated below, some important issues seem to be swept under the rug. They should be researched and clarified before the manuscript could be published. I cannot recommend the manuscript in its current form for publication in the Nature Communications.

1. "Very recently, it is shown that the magnetic skyrmion in the frustrated magnetic system has remarkable physical properties that renovate our understanding of the chiral magnetic object [52–59]."

The order of citations gives a wrong perspective. This field, arguably, started with Monte Carlo simulations showing the order-from-disorder stabilization of the skyrmion crystal in the triangular model with competing interactions [T. Okubo, S. Chung, and H. Kawamura, Phys.

Rev. Lett. 108, 017206 (2012). Phase diagram of an anisotropic frustrated magnet and properties of skyrmions with arbitrary vorticity and helicity have been discussed in Ref. [53]. Ref. [52], which is in the beginning of author's list, appeared later.

2. "The energies of a skyrmion with different helicity are degenerate in the absence of the DDI, but the degeneracy is resolved by the DDI."

Even in absence of dipole-dipole interactions the independence of skyrmion energy of the helicity angle is destroyed by the in-plane quartic magnetic anisotropy, which is allowed by symmetry of the square lattice of spins but not included into the model (1).

3. I have a number of remarks/questions concerning the treatment of dipole-dipole interactions.

(a) The effect of long-range dipole-dipole interactions strongly depends on linear dimensions of the magnetic film, e.g. the film thickness and the length-to-width ratio. Whether magnetodipolar interactions are strong or weak with respect to exchange interactions crucially depends on these parameters as well as on the value of the saturation magnetization. The question is whether dipole-dipole interactions are important in realistic films rather than in the lattice model considered in the manuscript. It should be possible to identify a dimensionless parameter (perhaps, more than one) characterizing the relative strength of the dipole-dipole interactions.

(b) Dipole-dipole interactions in ferromagnetic films with perpendicular magnetic anisotropy give rise to stripe domain patterns and, under an applied magnetic field, to arrays of magnetic bubbles which also have skyrmion topology. Should such patterns appear in frustrated magnets and, if so, how large is the period of stripe and bubble domains compared to the size of the lattices considered in the manuscript and the skyrmion radius?

(c) In general, spin ordering near edges of the nanostrips is different from that inside the strips. Since the authors do not mention any edge effects, one might conclude that they use periodic boundary conditions for spins, which in nanostructures with dipole-dipole interactions would be a wrong thing to do.

I did not find a discussion of these questions in the text.

4. Page 2, second column. "In this work, the values for the NN, NNN, and NNNN exchange interactions are given as $J_1 = 3$ meV, $J_2 = -0.8J_1$, and $J_3 = -1.2J_1$, respectively."

What determined this choice of parameters? Are there any materials with similar parameters to which the results obtained in the manuscript can be applied?

5. Phase diagram shown in Fig. 1a does not seem to include various spiral and multiply-periodic

states discussed in Refs. [52] and [53], which makes one think that the states shown in this phase diagram may not be global energy minima. What happened to the rich phase diagram of frustrated magnets? Do magnetodipolar interactions suppress spiral states? This has to be discussed.

6. Page 3, second column. "However the skyrmion with $Q = 4$ is unstable and split into two skyrmions with $Q = 2$. The helicity η changes during this splitting process from that of the initial state with $\eta = \pm \pi/2$ ".

It is not clear whether it makes sense to talk about the helicity of the $Q = -4$ skyrmion. Helicity of $Q = 1$ skyrmion (or rather skyrmion with vorticity $+1$) is defined as the angle between in-plane spin component and the radial direction. The energy of Dzyloshinskii-Moriya and dipole-dipole interactions strongly depends on this angle.

For skyrmions with Q different from $+1$, the angle between the in-plane spin vector and the radial direction varies along a contour encircling the skyrmion center. One can still define η by Eq.(6), but it does not have the same meaning as the helicity for the $Q = 1$ skyrmion. Therefore, in my opinion it does not have much sense to talk about change or conservation of helicity when the $Q = 4$ skyrmion splits into four $Q = 1$ skyrmions.

7. The results on the helicity-dependent motion of skyrmions driven by "low" electric currents (Figures 6c and 6d) are nice. I am surprised, however, that the authors do not provide any explanation of this effect, using e.g. an effective Thiele description of the center-of-mass dynamics (Refs. [52,54]). What determines the threshold value of the electric current? How important is magnetic frustration here and would this result apply to magnetic bubbles, the helicity of which is also determined by magnetodipolar interactions.

8. English can and should be improved on numerous occasions, see e.g. "...spontaneously form a bound state", "...the separation of the bound state", "high-profile topological object", "renovate our understanding", etc.

Reviewer #3 (Remarks to the Author):

Application of skyrmions for spintronic devices is a hot topic. In the manuscript, the authors reported that the helicity of skyrmions can be used as information carrier. They investigated numerically frustrated magnetic materials with including the dipole-dipole interaction (DDI) that was not included in the previous study, ref. 52. Remarkably, degeneracy of helicities is resolved by DDI. As a result, two helicities, which are of Bloch-type states, become the lowest energy

states. The major claim of the paper is that those Bloch-type states are used as information carrier in the binary memory: Two states can be distinguished by their dynamics under a weak driving current and one state can be converted to the other by applying a strong driving current.

The idea of using helicity as information carrier would be novel and of interest to others, and influence thinking in the field. The claim is appropriately discussed citing previous literature. The numerical simulation method appears to be reliable because it uses the Object Oriented MicroMagnetic Framework software, which is widely used in the field.

Basically I am positive to recommend publication of the manuscript in Nature Communications but there are several points to be clarified.

1. I wonder the value of J_3 is quite large. The value of $J_3 = -1.2 J_1$ is assumed throughout the paper. However, J_3 is the next-next-nearest-neighbor interaction. So, $|J_3|$ is much smaller than $|J_1|$ in general. Furthermore, J_3 is antiferromagnetic while J_1 is ferromagnetic. Therefore, the parameter choice is unreasonable.

2. The energy difference is scaled by DDI among skyrmion states with different helicities as shown in Fig. 4. That means we need to cool the device to temperatures lower than the energy scale of DDI. Therefore, the temperature would be on the order of 10 K. That is quite low.

3. In order to flip the helicity, a driving current is necessary which is on the order of 10^{11} A/m². This is quite large. The order is comparable to that required for domain wall motion. In other skyrmion systems, the order is five or six orders of magnitude smaller than this value.

Response to the Report of the First Reviewer

Reviewer #1 (Remarks to the Author):

This paper is a theoretical-computational study of skyrmion states in a frustrated magnet. Skyrmions are a subject of intense interest and have been found in numerous chiral magnets. Frustrated systems are another place where such skyrmions may arise as has been reported by other authors including some on this paper. Here the authors consider the effect of dipole-dipole interactions and examine which gives rise to a helicity. The authors use micromagnetic simulations to show that skyrmion states with different Q values can arise and then study various aspects of dynamics such as helicity locking and unlocking, annihilation and creation of bound states. They indicate how the dynamics of such states could be valuable for the creation of new kinds of skyrmion devices.

Overall the results are interesting and the dynamics of skyrmions in frustrated magnetism is not very well known. The authors find a rich variety of dynamical behaviors. I have some questions before making a conclusion either way.

Authors' Reply:

We appreciate the reviewer for noting our work to be interesting, and pointing out our results reveal a rich variety of dynamical behaviors of frustrated skyrmions. We also thank the reviewer for his/her valuable comments on the manuscript. Based on these comments, we have carried out a number of additional simulations as well as theoretical analysis, and revised the manuscript carefully and thoroughly. Besides, we have re-performed a literature study, and cited several references related to the frustrated materials as well as the experimental observations of skyrmions and antiskyrmions. Please find our detailed replies to the comments below.

Reviewer #1's Comment 1:

(1) The authors should go over the text as some of the wording is bit awkward; however, the paper is in general easy to follow.

Authors' Reply:

We have corrected grammatical mistakes, and checked the whole manuscript to eliminate other typos.

Reviewer #1's Comment 2:

(2) The authors do not spend much time on thermal effects. Thermal effects are known to be important for stabilizing skyrmions in chiral magnets, but how about in frustrated magnets? Additionally, could one have thermal unbinding of bound pairs?

Authors' Reply:

Although our present study is focused on the zero-temperature physics of magnetic skyrmions in frustrated systems with dipolar interactions, as the reviewer suggested, it would be insightful to explore the thermal effects and show some preliminary results. Following the reviewer's comments, we have studied the thermal effect on the motion dynamics of skyrmions in both cases of weak and strong driving, where the spin dynamics is solved by the stochastic Landau-Lifshitz-Gilbert equation (see Methods). It is found that the trajectory of a skyrmion fluctuates once the thermal effect is applied. Indeed, the stability of the skyrmion decreases with increasing amplitude of the thermal fluctuations. When the thermal effect is stronger than a certain threshold, the skyrmion becomes very unstable and could be destroyed. On the other hand, we have also investigated the thermal effect on the skyrmion-skyrmion bound state, that is, a bi-skyrmion with $Q = 2$. It is found that the bi-skyrmion shows Brownian motion behavior, of which the magnitude is proportional to the thermal effect. The stability of the bi-skyrmion decreases with increasing magnitude of the thermal fluctuations. We found the bi-skyrmion with $Q = 2$ could be collapsed to a skyrmion with $Q = 1$ under certain amplitude of the thermal fluctuations. Also, the bi-skyrmion or skyrmion could be annihilated when the thermal effect is stronger than a certain threshold magnitude. Interestingly, we indeed found that the thermal unbinding event of a bi-skyrmion happened before the annihilation of skyrmions at certain magnitude of the thermal fluctuations, which we believe is a meaningful event of frustrated skyrmions with higher topological charges. However, we think further exploration of thermal unbinding events is beyond the scope of our present manuscript as we mainly focus on the zero-temperature phenomena. Just like the reviewer pointed out, the zero-temperature dynamics of skyrmions in frustrated magnetics is not very well known at the moment, which should be studied at first.

We have mentioned the results related to the thermal effect in the main text and methods. The detailed results on the thermal effect are given in the supplementary document: see Supplementary Figure 18 and Supplementary Figure 24.

Reviewer #1's Comment 3:

(3) Experimentally what would one look for in these systems? For example, what signatures could be observed through direct imaging, transport or neutron scattering?

Authors' Reply:

In principle, the frustrated skyrmions can be observed experimentally through a number of techniques. The one of the most essential signatures of frustrated skyrmions is the helicity, which is a degree of freedom and could be used as information carriers. The helicity is determined by the in-plane spin texture of a skyrmion. Thus, for experimentalists, it is an important task to observe and identify the helicity of a frustrated skyrmion. Fortunately, there are a lot of published works demonstrating the possibility of observation and even control of skyrmion helicity. For examples, the Lorentz transmission electron microscopy (LTEM) can be used to directly observe the helicity of a nanoscale frustrated skyrmion [Nat. Commun. 3, 988 (2012); Nat. Nanotech. 8, 723 (2013); Nat. Commun. 5, 3198 (2014); Nat. Commun. 6, 8504 (2015); APL 109, 022402 (2016); PNAS 109, 8856, (2017)]. It is worth mentioning that the antiskyrmion, which has a quite different helicity to the skyrmion, has recently been observed by using the LTEM [arXiv:1703.01017]. The skyrmion bound pairs have also been observed by using the LTEM [Nat. Commun. 5, 3198 (2014); APL 109, 022402 (2016)]. Besides, the helicity of a skyrmion can be observed by using the differential phase contrast transmission electron microscopy (DPC STEM) technique [Sci. Adv. 2, e1501280 (2016); Sci. Rep. 6, 35880 (2016)]. On the other hand, a new experimental protocol for the identification of topological magnetic structures with different helicities, by soft X-ray spectroscopy, is proposed [Nat. Commun. 7, 13613 (2016)]. It is also worth mentioning that a measurable quantity associated with the skyrmion helicity is the toroidal moment [J. Phys.: Condens. Matter 20, 434203 (2008)], which has been discussed in Refs. [Nat. Commun. 6, 8275 (2015); PRB 93, 064430 (2016)].

Based on the reviewer's comment, we have added a brief discussion on the observation of frustrated skyrmions in the main text. In addition, we have cited more references related to the observations of signatures of skyrmions, antiskyrmions, and bound skyrmion states.

Reviewer #1's Comment 4:

(4) The skyrmion lattice structure generally looks disordered in these systems. Is this due to the frustrating effects or the equilibration time in the simulations, or is this a general feature one would expect in these types of skyrmions as opposed to skyrmions in DMI materials?

Authors' Reply:

The disordered distribution of frustrated skyrmions and antiskyrmions in our present study is due to the reason that our simulated samples have confined geometries and the open boundary conditions. Isolated skyrmions and antiskyrmions in the confined geometries may attract or repel each other, forming disordered skyrmion clusters, or skyrmion-antiskyrmion gas or liquid. As shown in Ref. [PRB 93, 064430 (2016)], the distribution of the skyrmion-antiskyrmion liquid is rather disordered. Also, we considered the presence of the dipolar interactions in our studied system, which could result in non-periodic spin textures in the confined geometries. In real-world experiments, the side length of the magnetic samples can be as large as several micrometers, in which skyrmion lattice or periodic spin textures could certainly be formed in certain parameter spaces as well as external conditions (temperature, applied field, etc.). Indeed, when the sample size is limited and the material parameters are in specific ranges, the lack of regular lattice structure of skyrmions can be observed as shown in recent experiments [Science 349, 283 (2015); Nat. Mater. 15, 501 (2016); Nat. Phys. 13, 162 (2017); Nat. Phys. 13, 170 (2017); Nat. Commun. 8, 15573 (2017)]. In the revision of the manuscript, we have carried out additional simulations for a range of J1-J2-J3 interactions, anisotropies, and applied fields. It can be seen that periodic spin textures can be formed in certain parameter space regions. On the other hand, we have re-simulated the K-H phase diagram by using the Landau-Lifshitz-Gilbert solver with enough long relaxation time, where we find the results are qualitatively same as that obtained by the conjugate gradient minimizer from random distributed spin configurations. The formation of skyrmion lattice should be a general feature for both frustrated magnets and magnets with Dzyaloshinskii-Moriya interactions, however, the formation of a skyrmion-antiskyrmion lattice would be an unusual feature of the frustrated magnets [Nat. Commun. 6, 8275 (2015)], as the nanoscale antiskyrmions are not energetically favorable in magnets with Dzyaloshinskii-Moriya interactions.

We have presented additional simulation results on the relaxation as well as new phase diagrams in the supplementary document: see Supplementary Figures 1-10. These new results are mentioned and discussed in the main text.

Reviewer #1's Comment 5:

(5) The authors discuss the dynamics; however, experimentally how would one detect this? Could one look at changes in the topological Hall effect?

Authors' Reply:

This comment is related to reviewer #1's comment 3. As discussed in our manuscript and Refs. [Nat. Commun. 6, 8275 (2015); PRB 93, 064430 (2016); Nat. Commun. 8, 14394 (2017)], the dynamics of a frustrated skyrmion is different from that of a skyrmion in conventional ferromagnetic system with Dzyaloshinskii-Moriya interactions. The current-induced motion of a frustrated skyrmion is coupled with its helicity dynamics. When the helicity is locked under weak driving, the frustrated skyrmion shows translational motion, while it shows rotational motion when the helicity is unlocked under strong driving. Either the translational or rotational motion of the frustrated skyrmion in thin-films or multilayers can be directly observed by the Lorentz transmission electron microscopy (LTEM). It is also possible to reveal the picosecond dynamics of a frustrated skyrmion using the direct time-resolved X-ray microscopy [Nat. Mat. 15, 501 (2016); Nat. Phys. 170, (2017)]. On the other hand, due to the skyrmion Hall effect, the frustrated skyrmions and antiskyrmions in constricted geometries will move toward and accumulate at opposite edges. Because the topological Hall effect is a manifestation of emergent magnetic field, where the Hall effect occurs in the presence of skyrmions or antiskyrmions, and depends on the position of isolated skyrmions or antiskyrmions [APL 108, 112401 (2016)], it would be possible to detect isolated skyrmions and antiskyrmions by utilizing the topological Hall effect, as suggested by Kanazawa et al. [PRB 91, 041122(R) (2015)]. It is noteworthy that the emerging magnetic field of a skyrmion is inversely proportional to the area of the skyrmion, which means the nanometer-scale frustrated skyrmions studied in our manuscript would be a good candidate for the measurement of topological Hall effect.

Based on the reviewer's comment, we have discussed the detection of frustrated skyrmions in the main text. In addition, we have cited some references related to the detection and observation of skyrmions.

**Reviewer #1's Comment 6:**

(6) What would be the typical lattice constant of these systems or the size of the skyrmions? Do the authors have any particular materials in mind or do experimentalists need to find a way to make J1-J3 materials in the right conditions to find these states?

Authors' Reply:

In our present manuscript, we employed a lattice constant of 0.4 nm, as the typical lattice constant of the magnetic spin system is usually below or around 1 nm. The size of the frustrated skyrmion with $Q = 1$ in our studied system is about 5 times of the lattice constant, where the skyrmion size is defined by the diameter of the circle of $m_z = 0$. Also, we have found that the skyrmion size increases with increasing topological number Q . The reason is that the skyrmion with a higher Q has a more complex in-plane spin texture, which is formed by more spins. In real experiments, under certain conditions or effects, for examples, small anisotropy, pinning potential effect, and thermal effect, the frustrated skyrmions could be much bigger than the lattice parameter of the spin system and decouple from the spin lattice. With respect to the materials and experimental conditions, competing J1-J2 or J1-J2-J3 interactions and a weak easy axis anisotropy are sufficient to host frustrated skyrmions. As already discussed in Refs. [PRL 108, 017206 (2012); Nat. Commun. 6, 8275 (2015)], the complex (incommensurate) spin textures resulting from the competing interactions have been observed in triangular magnets with transition metal ions, such as NiGa₂S₄ [Science 309, 1697 (2005); PRL 105, 037402 (2010)] and α -NaFeO₂ [PRB 76, 024420 (2007)], and in some dihalides, for example, Fe_xNi_{1-x}Br₂ [J. Phys. 43, 1283 (1982)]. The anisotropy of dihalides can be adjusted by chemical substitutions, for example, Fe_xNi_{1-x}Br₂ is an easy axis magnet for $x > 0.1$ [J. Solid State Chem. 59, 23 (1985)]. Also, it is noteworthy that in some J1-J2-J3 materials, the magnitude of the antiferromagnetic J3 interaction could be much larger than that of the ferromagnetic J1 interaction ($|J3|/J1 \approx 5$) [PRL 108, 017206 (2012)], which means the frustrated skyrmions or other topological spin textures may be ubiquitous in these magnets with strong competing and frustrated interactions. On the other hand, in addition to the J1-J2-J3 frustrated spin system studied in our manuscript, there are actually many more different frustrated (competing) systems and materials, for example, the materials having the Kitaev interactions [Sci. Rep. 6, 26750 (2016)], which may be used to study the dynamics of topological spin textures, such as skyrmions and vortices.

Based on the reviewer's comment, we have discussed the materials hosting frustrated skyrmions in the main text. We have also cited some references related to the materials with competing or frustrated spin interactions.

Response to the Report of the Second Reviewer

Reviewer #2 (Remarks to the Author):

The authors discuss static and dynamic properties of skyrmions in a frustrated square-lattice magnet with competing exchange interactions and long-range magnetodipolar interactions. The dipole-dipole interaction favors Bloch-type skyrmions (helicity angle equal plus/minus $\pi/2$), as is also the case for magnetic bubbles with skyrmion topology, and the authors explore the effect of this "helicity fixing" on the current-driven dynamics of skyrmions. They find, in particular, that the direction of motion excited by low electric currents depends on the skyrmion helicity, while stronger currents give rise to helicity rotation accompanying the translational motion of skyrmion. The latter effect is easy to understand, but the former one is a surprise to me. The authors suggest that these effects can be used to make new magnetic memory devices.

*I have mixed feelings about this work. **While some results look interesting, I am not happy with the overall quality of the manuscript. As is elaborated below, some important issues seem to be swept under the rug. They should be researched and clarified before the manuscript could be published.** I cannot recommend the manuscript in its current form for publication in the Nature Communications.*

Authors' Reply:

We thank the reviewer for careful reading of our manuscript, and for noting that our results are interesting. Following the suggestions of the reviewer, we have made considerable modification of the manuscript. First of all, we have presented a theoretical analysis of the locking-unlocking transition in the helicity dynamics based on the Thiele approach. We have derived the analytical formula for the transition point: see the reply to reviewer #2's comment 7 more in details. Furthermore, we have carried out a large number of additional simulations to enrich the results of the manuscript. Besides, we have re-performed a literature study, and cited several references related to the frustrated materials as well as the experimental observations of skyrmions. In specific, we have highlighted Refs. [Nat. Commun. 6, 8275 (2015); Nat. Commun. 8, 14394 (2017)] in several places of the revised manuscript, as the rich phase diagrams of frustrated magnets and some frustrated skyrmion dynamics have been well studied in these prior works. We address the points raised by the reviewer in details below. We hope that our revision resolves the reviewer's uneasiness of the overall quality of our original manuscript.

Reviewer #2's Comment 1:

1. *"Very recently, it is shown that the magnetic skyrmion in the frustrated magnetic system has remarkable physical properties that renovate our understanding of the chiral magnetic object [52-59]."*

The order of citations gives a wrong perspective. This field, arguably, started with Monte Carlo simulations showing the order-from-disorder stabilization of the skyrmion crystal in the triangular model with competing interactions [T. Okubo, S. Chung, and H. Kawamura, Phys. Rev. Lett. 108, 017206 (2012)]. Phase diagram of an anisotropic frustrated magnet and properties of skyrmions with arbitrary vorticity and helicity have been discussed in Ref. [53]. Ref. [52], which is in the beginning of author's list, appeared later.

Authors' Reply:

Following the reviewer's comment, we have updated the order of the mentioned references. In addition, we have rewritten the introduction and discussion parts to present more information on the frustrated spin systems and materials.

Reviewer #2's Comment 2:

2. "The energies of a skyrmion with different helicity are degenerate in the absence of the DDI, but the degeneracy is resolved by the DDI."

Even in absence of dipole-dipole interactions the independence of skyrmion energy of the helicity angle is destroyed by the in-plane quartic magnetic anisotropy, which is allowed by symmetry of the square lattice of spins but not included into the model (1).

Authors' Reply:

We have added a comment on the in-plane quartic magnetic anisotropy in the discussion part. We have also pointed out that the in-plane quartic magnetic anisotropy $K \sum_i [(m_i^x)^4 + (m_i^y)^4]$ allowed by symmetry of the underlying square lattice can give rise to a non-uniform rotation of spins [arXiv:1705.02874] and destroy the helicity degeneracy. Besides, we have cited a reference [arXiv:1705.02874] related to the above statement. However, the in-plane quartic magnetic anisotropy exists in a specific class of materials such as cobalt ferrites and is negligible in ordinary magnets. Especially, there is no experimental report on the existence of the in-plane quartic magnetic anisotropy in frustrated magnets. The inclusion of this anisotropy is beyond the scope of this paper.

Reviewer #2's Comment 3.1:

3. I have a number of remarks/questions concerning the treatment of dipole-dipole interactions.

(a) The effect of long-range dipole-dipole interactions strongly depends on linear dimensions of the magnetic film, e.g. the film thickness and the length-to-width ratio. Whether magnetodipolar interactions are strong or weak with respect to exchange interactions crucially depends on these parameters as well as on the value of the saturation magnetization. The question is whether dipole-dipole interactions are important in realistic films rather than in the lattice model considered in the manuscript. It should be possible to identify a dimensionless parameter (perhaps, more than one) characterizing the relative strength of the dipole-dipole interactions.

Authors' Reply:

We agree with the reviewer that the effect of dipolar interactions depends on the geometry of the magnetic sample, as well as the saturation magnetization. Hence, we have investigated the effect of the dipolar interactions by simulating the magnetic samples with different length-to-width ratio, thickness, and saturation magnetization, where a skyrmion with topological charge $Q = 1$ and helicity $\eta = \pi/2$ is relaxed at the center of the magnetic sample (corresponding to the case shown in Fig. 3 of the main text). It is found that the DDI energy is monotonically proportional to the length-to-width ratio and the thickness. The DDI energy also increases with increasing saturation magnetization. On the other hand, the J_1 , J_2 , and J_3 exchange interaction energies are independent of the length-to-width ratio and saturation magnetization, and slightly increase with the thickness. The energy difference between the DDI and exchange interactions increases with increasing length-to-width ratio, thickness, and saturation magnetization. Hence, it can be seen that the DDI will have a strong effect in thick films with large saturation magnetization. Recently, it has been a trend to fabricate the skyrmion samples or devices in a multilayer fashion. It can be expected that DDI will be a more important factor in these multilayer structures, which may be developed for future applications, such as the multilayer skyrmion racetrack memory. In other words, we believe that the applications based on the manipulation of frustrated skyrmions are more possible to be implemented in multilayer structures instead of thin films in the near future, where the effect of DDI will play an important role.

We have mentioned the results related to the effect of DDI as functions of the geometric parameters (length-to-width ratio and thickness) and material parameter (saturation magnetization) in the main text. The detailed results about the effect of DDI are given in the supplementary document: see Supplementary Figure 13.

Reviewer #2's Comment 3.2:

(b) Dipole-dipole interactions in ferromagnetic films with perpendicular magnetic anisotropy give rise to stripe domain patterns and, under an applied magnetic field, to arrays of magnetic bubbles, which also have skyrmion topology. Should such patterns appear in frustrated magnets and, if so, how large is the period of stripe and bubble domains compared to the size of the lattices considered in the manuscript and the skyrmion radius?

Authors' Reply:

We have carried out a number of simulations and obtained several phase diagrams as functions of J_2 - J_3 exchange interactions, anisotropy, and applied magnetic field. In these phase diagrams, we found that the stripe domain patterns exist in certain parameter space regions. We also found skyrmions with a quite large size in the given system. However, it should be noted that the side length of our simulated samples is only of several tens of nanometers, which is limited by the computational ability of our workstations, thus it is impossible to obtain micrometer-scale bubbles in our present study. As can be measured from some results shown in the phase diagrams (see Supplementary Figures 6-7), the period of the stripe domains is typically 5 times the lattice constant. The size of skyrmions varies with the parameters, and depends on its topological charge and/or helicity. For a skyrmion with $Q = 1$ and $\eta = \pi/2$, its diameter is also about 5 times the lattice constant.

We have mentioned the size of skyrmions, and the formation of stripe domains as well as their period size in the main text. The detailed results of the new obtained phase diagrams are given in the supplementary document: see Supplementary Figures 3-6.

Reviewer #2's Comment 3.3:

(c) In general, spin ordering near edges of the nanostrips is different from that inside it the strips. Since the authors do not mention any edge effects, one might conclude that they use periodic boundary conditions for spins, which in nanostructures with dipole-dipole interactions would be a wrong thing to do. I did not find a discussion of these questions in the text.

Authors' Reply:

In our present study, we have employed the open boundary conditions in all simulations. At the same time, all parameters are defined in a spatial uniform manner in our simulations. Hence, in our results shown in the main text where the background magnetization is perpendicular to the film plane, the spin ordering near the edges of the sample is also perpendicular to the film plane. As pointed out in Ref. [Nat. Commun. 8, 14394 (2017)], the competing spin interactions in frustrated magnets do not necessarily induce the spin tilts on the edge similar to that observed in magnets with Dzyaloshinskii-Moriya interactions. Indeed, as also pointed out in Ref. [Nat. Commun. 8, 14394 (2017)], the exchange interactions and magnetic anisotropies near edges can be different from those in the interior of the magnetic sample, which will result in tilted spin ordering near the edge. Given that our present work focuses on the skyrmion dynamics in the interior of magnetic samples, we did not study the edge effects or edge states, which could be induced by in-plane anisotropy, for example. However, we have performed some simulations as functions of out-of-plane and in-plane applied fields. From the results, we can see that the edge spin configuration could still be different from the bulk spin configuration at certain amplitudes of the applied field before the sample is fully polarized.

We have added these simulation results about the edge profile at different applied fields to the supplementary document: see Supplementary Figure 11. Besides, we have briefly mentioned the point of edge states in the main text and highlighted the work [Nat. Commun. 8, 14394 (2017)], in which the edge states in a frustrated magnet have been systematically and carefully studied.

Reviewer #2's Comment 4:

4. Page 2, second column. "In this work, the values for the NN, NNN, and NNNN exchange interactions are given as $J_1 = 3 \text{ meV}$, $J_2 = -0.8J_1$, and $J_3 = -1.2J_1$, respectively."

What determined this choice of parameters? Are there any materials with similar parameters to which the results obtained in the manuscript can be applied?

Authors' Reply:

As our present work is a pure theoretical-computational study, the values for parameters J_1 , J_2 , and J_3 are chosen for stabilizing isolated skyrmions and antiskyrmions at creation conditions of the anisotropy and applied field. In order to demonstrate that the parameters used in our simulations are not specific and the skyrmions and antiskyrmions can be formed for a wide range of parameters, we have performed a lot of simulations to obtain the J_1 - J_2 - J_3 phase diagrams, where skyrmions and antiskyrmions are found to exist in certain parameter space regions. With respect to the materials, please refer to the reply to the reviewer #1's comment 6. It is worth mentioning that, as shown in Ref. [PRL 108, 017206 (2012)], in some J_1 - J_2 - J_3 materials the magnitude of the antiferromagnetic J_3 interaction could be much larger than that of the ferromagnetic J_1 interaction ($|J_3|/J_1 \approx 5$), which means that the frustrated skyrmions or other topological spin textures may be ubiquitous in these magnets with strong competing and frustrated interactions.

We have presented the new obtained phase diagrams as functions of exchange interactions in the supplementary document: see Supplementary Figures 3-6. Also, we have mentioned and highlighted in the main text that the rich phase diagrams of the J_1 - J_2 frustrated magnet have been well studied in Ref. [Nat. Commun. 6, 8275 (2015)].

Reviewer #2's Comment 5:

5. Phase diagram shown in Fig. 1a does not seem to include various spiral and multiply-periodic states discussed in Refs. [52] and [53], which makes one think that the states shown in this phase diagram may not be global energy minima. What happened to the rich phase diagram of frustrated magnets? Do magnetodipolar interactions suppress spiral states? This has to be discussed.

Authors' Reply:

This comment is related to reviewer #1's comment 4. First, in our present study, we only considered the open boundary conditions, which means it is impossible to obtain perfect periodic spin textures shown in Refs. [Nat. Commun. 6, 8275 (2015); PRB 93, 064430 (2016)]. Second, the dipolar interaction considered in our work is a long-range interaction; it is determined by the spin configuration of the whole sample and, in turn, influences the spin configuration. That is to say, the dipolar interaction is part of the factors that lead to the complex spin textures. On the other hand, the formation of either spiral states or skyrmions also depends on the parameters such as the strengths of J_2 and J_3 exchange interactions, and values of anisotropy and applied fields. In order to find the spiral states, we have carried out a number of simulations, and obtained more phase diagrams as functions of J_2 and J_3 exchange interactions, anisotropy, and applied field. It is found that the spiral states can be formed in a wide range of parameter space [see Supplementary Figures 5-6]. Besides, we also found that some $2q$ -states and $2q'$ -states are formed at certain parameters [see Supplementary Figures 3-4]. Indeed, in order to check whether or not the dipolar interactions will suppress the formation of spiral states, we have performed a number of simulations in the absence of the dipolar interactions, and obtained several phase diagrams [see Supplementary Figures 7-10]. In these phase diagrams, we can still find that the spiral states and skyrmion states can be formed, although the parameter conditions may not be exactly identical to those required for the formation of spiral states and skyrmion states in the presence of the dipolar interactions.

We have presented the new obtained phase diagrams in presence and absence of the dipolar interactions in the supplementary document: see Supplementary Figures 3-10, where spiral states, skyrmion states, and other types of spin textures can be found. In addition, we have mentioned and highlighted in the main text that the rich phase diagrams of the J_1 - J_2 frustrated magnet have been well studied in Ref. [Nat. Commun. 6, 8275 (2015)], in which several types of spin textures are discussed.

Reviewer #2's Comment 6:

6. Page 3, second column. "However the skyrmion with $Q = 4$ is unstable and split into two skyrmions with $Q = 2$. The helicity η changes during this splitting process from that of the initial state with $\eta = \pm \pi/2$ ".

It is not clear whether it makes sense to talk about the helicity of the $Q = -4$ skyrmion. Helicity of $Q = 1$ skyrmion (or rather skyrmion with vorticity $+1$) is defined as the angle between in-plane spin component and the radial direction. The energy of Dzyaloshinskii-Moriya and dipole-dipole interactions strongly depends on this angle.

For skyrmions with Q different from $+1$, the angle between the in-plane spin vector and the radial direction varies along a contour encircling the skyrmion center. One can still define η by Eq.(6), but it does not have the same meaning as the helicity for the $Q = 1$ skyrmion. Therefore, in my opinion it does not have much sense to talk about change or conservation of helicity when the $Q = 4$ skyrmion splits into four $Q = 1$ skyrmions.

Authors' Reply:

We agree with the reviewer that it is not meaningful and necessary to talk about the charge and conservation of helicity for the case of $Q = 4$. We have rephrased the mentioned discussion on the skyrmion with $Q = 4$. Indeed, in our present study, we are mainly focusing on the skyrmions with $Q = 1$ and $Q = 2$.

Reviewer #2's Comment 7:

7. The results on the helicity-dependent motion of skyrmions driven by “low” electric currents (Figures 6c and 6d) are nice. I am surprised, however, that the authors do not provide any explanation of this effect, using e.g. an effective Thiele description of the center-of-mass dynamics (Refs. [52,54]). What determines the threshold value of the electric current? How important is magnetic frustration here and would this result apply to magnetic bubbles, the helicity of which is also determined by magnetodipolar interactions.

Authors' Reply:

The comment consists of two parts. The answer to the first part reads as follows:

We have newly added the Thiele equation analysis in the revised manuscript, which describes the center-of-mass dynamics of a skyrmion. We have also provided the Thiele equation for the helicity-dependent motion of a skyrmion (see Methods). The Thiele equations in the absence of the DDI were studied in Refs. [Nat. Commun. 8, 14394 (2017); PRB 93, 064430 (2016)]. By adding an effective potential term due to the DDI, we have shown that the effective Thiele equation becomes identical to a forced oscillation problem of a pendulum in the presence of friction, where the force is provided by the applied driving current while the friction by the Gilbert damping. We find that when the force is less than the critical value the pendulum stops due to the friction, which corresponds to that a straight motion of the skyrmion with a fixed helicity. On the other hand, when the force is larger than the critical value, the pendulum rotates around the rotational center, which corresponds to that the helicity rotates. Furthermore, the critical value of the driving current density is determined by the potential barrier between Bloch-type and Néel-type skyrmions induced by the DDI, as well as the efficiency of the spin-transfer torque acting on the skyrmion.

We have added the analysis based on the effective Thiele equations in the Methods section of the main text. We have also discussed the determination of the threshold current density in the main text, and presented the related results in the supplementary document: see Supplementary Figures 14-15 and Supplementary Figure 17.

The answer to the second part reads as follows:

Magnetic bubbles usually have a very large size in the scale of micrometer. Furthermore, their circular domain wall structure is quite complicated. Their skyrmion number could be very large or zero, where the skyrmion number may be varied during the motion. For example, as shown in our recent work on the current-induced bubble motion [JMMM, in press, doi:10.1016/j.jmmm.2017.04.074 (2017)], the helicity of magnetic bubbles in ferromagnets may not be fixed and could fluctuate around an average value. Especially, one may be unable to displace magnetic bubbles with $Q = 0$ by the spin Hall torque [Science 349, 283 (2015)]. As a result, it is difficult to identify and control the exact spin configuration of the bubble's circular domain wall. Indeed, there is no report on the binary memory and current-induced switching with the use of magnetic bubbles.

Reviewer #2's Comment 8:

8. English can and should be improved on numerous occasions, see e.g. "...spontaneously form a bound state", "...the separation of the bound state", "high-profile topological object", "renovate our understanding", etc.

Authors' Reply:

We have corrected grammatical mistakes, and checked the whole manuscript to eliminate other typos.

Response to the Report of the Third Reviewer

Reviewer #3 (Remarks to the Author):

Application of skyrmions for spintronic devices is a hot topic. In the manuscript, the authors reported that the helicity of skyrmions can be used as information carrier. They investigated numerically frustrated magnetic materials with including the dipole-dipole interaction (DDI) that was not included in the previous study, ref. 52. Remarkably, degeneracy of helicity is resolved by DDI. As a result, two helicity states, which are of Bloch-type states, become the lowest energy states. The major claim of the paper is that those Bloch-type states are used as information carrier in the binary memory: Two states can be distinguished by their dynamics under a weak driving current and one state can be converted to the other by applying a strong driving current.

The idea of using helicity as information carrier would be novel and of interest to others, and influence thinking in the field. The claim is appropriately discussed citing previous literature. The numerical simulation method appears to be reliable because it uses the Object Oriented MicroMagnetic Framework software, which is widely used in the field.

Basically I am positive to recommend publication of the manuscript in Nature Communications but there are several points to be clarified.

Authors' Reply:

We thank the reviewer for careful reading of our manuscript. We also appreciate the reviewer for his/her very positive comments on our work. Based on the useful comments raised by the reviewer, we have performed a lot of additional simulations and revised the manuscript carefully and thoroughly. Moreover, we have re-performed a literature study, and cited several references related to the frustrated materials as well as the experimental observations of skyrmions. We address the points raised by the reviewer in details below.

Reviewer #3's Comment 1:

1. I wonder the value of J_3 is quite large. The value of $J_3 = -1.2 J_1$ is assumed throughout the paper. However, J_3 is the next-next-nearest-neighbor interaction. So, $|J_3|$ is much smaller than $|J_1|$ in general. Furthermore, J_3 is antiferromagnetic while J_1 is ferromagnetic. Therefore, the parameter choice is unreasonable.

Authors' Reply:

This comment is related to reviewer #2's comment 4. As our present work is a pure theoretical-computational study, the values for parameters J_1 , J_2 , and J_3 are chosen for stabilizing isolated skyrmions and antiskyrmions at creation conditions of the anisotropy and applied field. In order to demonstrate that the parameters used in our simulations are not specific and that the skyrmions and antiskyrmions can be formed at a wide range of parameters, we have performed a lot of simulations to obtain the J_1 - J_2 - J_3 phase diagrams, where skyrmions and antiskyrmions are found to exist in certain parameter space regions. With respect to the materials, please refer to the reply to the reviewer #1's comment 6. It should be noted that, as shown in Ref. [PRL 108, 017206 (2012)], in some J_1 - J_2 - J_3 materials the magnitude of the antiferromagnetic J_3 interaction could be much larger than that of the ferromagnetic J_1 interaction ($|J_3|/J_1 \approx 5$), which means that the frustrated skyrmions or other topological spin textures may be ubiquitous in these magnets with strong competing and frustrated interactions.

We have presented the new obtained phase diagrams as functions of exchange interactions in the supplementary document: see Supplementary Figures 3-6. Also, we have mentioned and cited some references related to the parameter settings and frustrated magnetic materials.

Reviewer #3's Comment 2:

2. The energy difference is scaled by DDI among skyrmion states with different helicities as shown in Fig. 4. That means we need to cool the device to temperatures lower than the energy scale of DDI. Therefore, the temperature would be on the order of 10 K. That is quite low.

Authors' Reply:

This comment is related to reviewer #1's comment 2. First, just like the reviewer #1 pointed out, the zero-temperature dynamics of skyrmions in frustrated magnetics is not very well known at the moment, which should be studied at first. Indeed, in the research of magnetic skyrmions, we can see that early experimental studies are also focused on low-temperature physics, for example, in Ref. [Science 341, 636 (2013)] the temperature is around 5 K. Hence, it would be reasonable that our present study can only focus on the zero-temperature physics of magnetic skyrmions in frustrated systems with dipolar interactions. However, as the reviewer #1 suggested, it would be insightful to explore the thermal effects and show some preliminary results and conclusions. Following the reviewer's comments, we have studied the thermal effect on the motion dynamics of skyrmions in both cases of weak and strong driving, where the spin dynamics is solved by the stochastic Landau-Lifshitz-Gilbert equation. It is found that the trajectory of a skyrmion fluctuates once the thermal effect is applied. Indeed, the stability of the skyrmion decreases with increasing amplitude of the thermal fluctuations. When the thermal effect is stronger than a certain threshold, the skyrmion is very unstable and could be destroyed.

We have mentioned the results related to the thermal effect in the main text and methods. The detailed results on the thermal effect are given in the supplementary document: see Supplementary Figure 18 and Supplementary Figure 24.

Reviewer #3's Comment 3:

3. In order to flip the helicity, a driving current is necessary which is on the order of 10^{11} A/m². This is quite large. The order is comparable to that required for domain wall motion. In other skyrmion systems, the order is five or six orders of magnitude smaller than this value.

Authors' Reply:

This comment is related to reviewer #2's comment 7. In particular, we have presented an analytic formula of the critical current density in the main text, of which the details are given in the methods. We found the critical value of the driving current density is determined by the potential barrier between Bloch-type and Néel-type skyrmions induced by the DDI, as well as the efficiency of the spin-transfer torque acting on the skyrmion. The values of the potential barrier and driving force efficiency depend on the spin configuration of the skyrmion. Since the spin configuration is determined by parameters such as the frustrated exchange interactions, anisotropy, and applied external field, the critical current density can thus be adjusted by controlling these parameters in experiments. Indeed, we have performed additional simulations to investigate the relations between the critical current density required to induce the helicity unlocking event and the parameters, such as J₂, J₃ exchanges interactions, anisotropy, and applied field. It is found that the threshold current density is very sensitive and almost proportional to the values of the J₂, J₃ exchange interactions, anisotropy, and applied field. It is found that the magnetic frustration is very important to the determination of the threshold current density, for example, the threshold current density can significantly decrease from about 130×10^{10} A m⁻² to about 50×10^{10} A m⁻² when J₂ decreases from -0.7 to -0.9 (in units of J₁ = 1). On the other hand, it is worth mentioning that pinning potentials in real-world materials, such as pinning sites and impurities, will influence the dynamics of isolated skyrmions in real experiments. Thus, the current density required to drive the skyrmion into a high-speed motion could be 40×10^{10} A m⁻² [Nat. Phys. 13, 170 (2017)], which is on the same order of magnitude of the current density used in our simulations.

We have discussed the dependence of threshold current density on parameters in the main text, and presented the related results in the supplementary document: see Supplementary Figures 14-15 and Supplementary Figure 17.

Reviewers' comments:

Reviewer #1 (Remarks to the Author):

I had previously reviewed this paper and find that the authors have satisfactorily respond to the issues I have raised. They have also made some changes to the text to address the points raised by the second referee some of which overlapped with me. I now think the paper is ready for publication.

Reviewer #2 (Remarks to the Author):

The revised manuscript is a considerable improvement on the previous text. I have only two remarks left.

1. Figure 1 in the main text as well as the Supplementary Figures 1-10 should not be called phase diagrams. This is clear from Figure 1b showing a collection of randomly placed skyrmions and antiskyrmions with different topological numbers. Such a state cannot possibly be the global energy minimum. Rather, the spin configurations shown in the blocks of these "phase diagrams" represent typical metastable states at which the energy minimization routine stops. To obtain the real phase diagram, one would have to repeat the minimization many times starting from different initial configurations and to use annealing techniques. Furthermore, one can only talk about phases and phase transitions in the thermodynamic limit (i.e. in the limit of infinitely large lattice). I suggest using the name phase diagram in quotes ("phase diagram") or to refer to these figures as collections of typical metastable states.

2. I am not quite satisfied with the authors' reply to my remark concerning the relative importance of magneto-dipolar interactions on the aspect ratio and lattice size. In particular, from the caption of the Supplementary Fig. 13 it is not clear what energies are plotted there. My guess is that the authors plot the exchange, anisotropy and dipole-dipole energies of the whole lattice containing one skyrmion. If this is the case, such a plot would not be very informative, because the dipole-dipole and exchange energies clearly scale differently with the aspect ratio. A plot of all these contributions to the skyrmion energy, i.e. the difference between the energy of the system with the skyrmion and without it, would be more meaningful.

Reviewer #3 (Remarks to the Author):

The authors replied to most of the referees' criticisms properly, and carried out numerical simulations required to support their claims.

However, the logic about the large value of J_3 is misleading. The paper, PRL 108, 017206 (2012), cited in authors' reply to Reviewer #1's Comment 6, Reviewer #2's Comment 4, and my comment 1, is a theoretical paper. It does not "show" $|J_3|$ can be much larger than $|J_1|$. Possible candidate is NiGa₂S₄: A large $|J_3/J_1|$ value is suggested in PRL 105, 037402 (2010) from the analysis of the spin wave observed by the neutron scattering.

If $|J_3/J_1|$ is large in NiGa₂S₄, it would be a special property of the triangular lattice and wave function overlaps in this compound. I do not understand why the authors take the square lattice. A natural choice would be the triangular lattice if one assumes large values of $|J_3/J_1|$.

A phase diagram on the J_2 - J_3 plane is shown in Supplementary Figure 3. But there is no discussion about the plausible range of J_2 and J_3 either in the main text and in the supplementary material.

A non-trivial thing can happen when some parameter takes a large value. Even if it is quite attractive, one must examine whether assumed parameter values are realistic or not.

If the authors clarify this point, I recommend publication of the manuscript in Nature Communications.

Response to the Report of the First Reviewer

Reviewer #1 (Remarks to the Author):

*I had previously reviewed this paper and find that the authors have satisfactorily respond to the issues I have raised. They have also made some changes to the text to address the points raised by the second referee some of which overlapped with me. **I now think the paper is ready for publication.***

Authors' Reply:

We appreciate the reviewer for carefully reviewing our manuscript and providing useful comments, which help us to improve the manuscript.

Response to the Report of the Second Reviewer

Reviewer #2 (Remarks to the Author):

The revised manuscript is a considerable improvement on the previous text. I have only two remarks left.

Authors' Reply:

We thank the reviewer for providing additional useful comments. Following the reviewer's comments, we have performed new simulations, updated a figure in the supplementary document, and revised the main text. We address the points raised by the reviewer in details below.

Reviewer #2's Comment 1:

1. Figure 1 in the main text as well as the Supplementary Figures 1-10 should not be called phase diagrams. This is clear from Figure 1b showing a collection of randomly placed skyrmions and antiskyrmions with different topological numbers. Such a state cannot possibly be the global energy minimum. Rather, the spin configurations shown in the blocks of these "phase diagrams" represent typical metastable states at which the energy minimization routine stops. To obtain the real phase diagram, one would have to repeat the minimization many times starting from different initial configurations and to use annealing techniques. Furthermore, one can only talk about phases and phase transitions in the thermodynamic limit (i.e. in the limit of infinitely large lattice). I suggest using the name phase diagram in quotes ("phase diagram") or to refer to these figures as collections of typical metastable states.

Authors' Reply:

We agree with the reviewer that the magnetic spin textures obtained by our relaxation simulations in the framework of micromagnetics are actually metastable states, *i.e.* the local energy minimum solutions. However, since it is not allowed to use quote marks in the text of *Nature Communications*, we have replaced "phase diagram" with "typical metastable states" in the main text and in the supplementary document.

Reviewer #2's Comment 2:

2. I am not quite satisfied with the authors' reply to my remark concerning the relative importance of magneto-dipolar interactions on the aspect ratio and lattice size. In particular, from the caption of the Supplementary Fig. 13 it is not clear what energies are plotted there. My guess is that the authors plot the exchange, anisotropy and dipole-dipole energies of the whole lattice containing one skyrmion. If this is the case, such a plot would not be very informative, because the dipole-dipole and exchange energies clearly scale differently with the aspect ratio. A plot of all these contributions to the skyrmion energy, i.e. the difference

Thursday, August 3, 2017

between the energy of the system with the skyrmion and without it, would be more meaningful.

Authors' Reply:

We thank the reviewer for pointing out the problem in Supplementary Figure 13. The reviewer is correct that in this figure, we plot the exchange, anisotropy and dipole-dipole energies of the whole lattice containing one typical ground-state skyrmion with $Q = 1$ and $\eta = \pi/2$. We agree with the reviewer such a plot is not informative as the dipole-dipole energy certainly depends on the geometry. Hence, we have re-plotted Supplementary Figure 13 by showing the energy difference between the systems with and without the skyrmion. We also improved the figure caption to provide more information regarding the calculations. Accordingly, we have updated Figure 3 and Figure 4 in the main text.

Response to the Report of the Third Reviewer

Reviewer #3 (Remarks to the Author):

The authors replied to most of the referees' criticisms properly, and carried out numerical simulations required to support their claims. However, the logic about the large value of J_3 is misleading. The paper, PRL 108, 017206 (2012), cited in authors' reply to Reviewer #1's Comment 6, Reviewer #2's Comment 4, and my comment 1, is a theoretical paper. It does not "show" $|J_3|$ can be much larger than $|J_1|$. Possible candidate is NiGa₂S₄: A large $|J_3/J_1|$ value is suggested in PRL 105, 037402 (2010) from the analysis of the spin wave observed by the neutron scattering. If $|J_3/J_1|$ is large in NiGa₂S₄, it would be a special property of the triangular lattice and wave function overlaps in this compound. I do not understand why the authors take the square lattice. A natural choice would be the triangular lattice if one assumes large values of $|J_3/J_1|$. A phase diagram on the J_2 - J_3 plane is shown in Supplementary Figure 3. But there is no discussion about the plausible range of J_2 and J_3 either in the main text or in the supplementary material. A non-trivial thing can happen when some parameter takes a large value. Even if it is quite attractive, one must examine whether assumed parameter values are realistic or not. If the authors clarify this point, I recommend publication of the manuscript in Nature Communications.

Authors' Reply:

We thank the reviewer for his/her very useful comments and those references. First, we agree with the reviewer that $|J_3/J_1| = 5$ is a possible value suggested by the mean field theory based on some experimental facts [Science 309, 1697 (2005); J. Phys. Soc. Jpn. 79, 011003 (2010)]. In the revised manuscript and the supplementary information, we have mentioned that $|J_3|$ is almost three times larger than $|J_1|$ in NiGa₂S₄ in the experimental work [PRL 105, 037402 (2010)], which is a special property of the triangular lattice and wave function overlaps. However, due to the limitation of our finite-difference simulation software, we are only able to consider the underlying square lattice. Nevertheless, as shown in [PRB 93, 064430 (2016)], both the square lattice and triangular lattice are possible candidates for hosting frustrated skyrmions, and worth investigating. In order to clarify the assumed relation between J_2 and J_3 , we have added new discussions to the main text as well as the figure caption of Supplementary Figure 3. We hope the reviewer will be satisfied with the revised manuscript.

Reviewers' Comments:

Reviewer #2 (Remarks to the Author):

I am happy with the changes made by the authors, especially with the Supplementary Fig. 13, where they now plot the contributions of various interactions to the skyrmion energy rather than to the total energy of the film. However, the conclusion made on page 4 in the main text,

"We show in Supplementary Fig. 13 that the DDI energy ... decreases with increasing thickness 227 of the sample. ... Thus, it can be expected that the DDI effect will be 229 more significant in thick samples with large MS."

seems unjustified to me.

The decrease of the negative DDI energy (i.e. the increase of its magnitude) shown in Fig. 13b is largely related to the fact that the energy of the skyrmion tube is proportional to the number of layers. Naturally, the contributions of other interactions also scale with the number of layers, as is clear from this figure. Thus, if one would like to make conclusions about the relative importance of the DDI, one should plot all the contributions to skyrmion energy per one layer. Even better, one can plot their ratio to the total skyrmion energy as a function of film thickness.

I have no further comments.

Reviewer #3 (Remarks to the Author):

The authors properly replied to my criticism and Reviewer #2's comments. I also found that the manuscript and Supplementary Information were satisfactory revised. Now I recommend publication of the manuscript in Nature Communications.

Response to the Report of the Second Reviewer

Reviewer #2 (Remarks to the Author):

I am happy with the changes made by the authors, especially with the Supplementary Fig. 13, where they now plot the contributions of various interactions to the skyrmion energy rather than to the total energy of the film. However, the conclusion made on page 4 in the main text, "We show in Supplementary Fig. 13 that the DDI energy ... decreases with increasing thickness of the sample. ... Thus, it can be expected that the DDI effect will be more significant in thick samples with large MS." seems unjustified to me.

The decrease of the negative DDI energy (i.e. the increase of its magnitude) shown in Fig. 13b is largely related to the fact that the energy of the skyrmion tube is proportional to the number of layers. Naturally, the contributions of other interactions also scale with the number of layers, as is clear from this figure. Thus, if one would like to make conclusions about the relative importance of the DDI, one should plot all the contributions to skyrmion energy per one layer. Even better, one can plot their ratio to the total skyrmion energy as a function of film thickness.

I have no further comments.

Authors' Reply:

We thank the reviewer for providing additional useful comments. Following the reviewer's comments, we have updated Supplementary Figure 13 to show all the energy contributions to the total skyrmion energy ratio as a function of film thickness, and accordingly revised the main text.

Response to the Report of the Third Reviewer

Reviewer #3 (Remarks to the Author):

The authors properly replied to my criticism and Reviewer #2's comments. I also found that the manuscript and Supplementary Information were satisfactory revised. Now I recommend publication of the manuscript in Nature Communications.

Authors' Reply:

We appreciate the reviewer for carefully reviewing our manuscript and Supplementary Information, and for recommending the publication of our work.